# Redox-dependent Igfbp2 signaling controls Brca1 DNA damage response to govern neural stem cell fate

Weam S. Shahin [1], Shima O. Ebed[1], Scott R. Tyler [1], Branko Miljkovic[1], Soon H. Choi [1], Yulong Zhang[1], Weihong Zhou[1], Idil A. Evans[1], Charles Yeaman[1] & John F. Engelhardt [1] ✉

Neural stem cell (NSC) maintenance and functions are regulated by reactive oxygen species (ROS). However, the mechanisms by which ROS control NSC behavior remain unclear. Here we report that ROS-dependent Igfbp2 signaling controls DNA repair pathways which balance NSC self-renewal and lineage commitment. *Ncf1* or *Igfbp2* deficiency constrains NSCs to a self-renewing state and prevents neurosphere formation. Ncf1-dependent oxidation of Igfbp2 promotes neurogenesis by NSCs in vitro and in vivo while repressing Brca1 DNA damage response genes and inducing DNA double-strand breaks (DDSBs). By contrast, *Ncf1*[−/−] and *Igfbp2*[−/−] NSCs favor the formation of oligodendrocytes in vitro and in vivo. Notably, transient repression of Brca1 DNA repair pathway genes induces DDSBs and is sufficient to rescue the ability of *Ncf1*[−/−] and *Igfbp2*[−/−] NSCs to lineage-commit to form neurospheres and neurons. NSC lineage commitment is dependent on the oxidizable cysteine-43 residue of Igfbp2. Our study highlights the role of DNA damage/repair in orchestrating NSC fate decisions downstream of redox-regulated Igfbp2.

Self-renewal, differentiation, and survival of tissue-specific stem cells (TSSCs) are regulated by reactive oxygen species (ROS)[1–5]. However, the mechanisms by which homeostatic and disease-associated changes in ROS influence fates of TSSCs in particular organs, including the brain, remain poorly defined. Neural stem cells (NSCs) are particularly sensitive to changes in both intracellular and extracellular redox states at homeostasis and following injury[6–8], and the NADPH oxidase complex (Nox/Ncf1) has been implicated as a major source of ROS that impact proliferation and differentiation of NSCs[6,9,10]. Nox2, the most studied Nox, transfers an electron from NADPH to molecular oxygen to generate superoxide. Thus, this pathway is extremely sensitive to oxygenation status and cellular metabolism, which can rapidly change following infarct. Superoxide can then be rapidly converted to hydrogen peroxide in the local environment through intracellular and extracellular superoxide dismutase enzymes, or spontaneous dismutation. Activation of Nox2 requires recruitment of cytosolic subunits

including neutrophil cytosolic factor 1 (Ncf1/p47phox) and Rac1 to membrane-bound components (p22phox and gp91phox)[11].

Multiple isoforms of Nox (1–4) are expressed in the brain where they appear to regulate NSC proliferation[12]. For example, ROS generated by Nox2 stimulates proliferation of neuronal progenitor cells (NPCs) and promotes neurogenesis in the adult midbrain[13]. In addition, ROS act to promote proliferation of adult hippocampal progenitor cells following Fgf stimulation[6]. Similarly, p53 temporally regulates neurogenesis by fine-tuning of cellular ROS levels in NSCs[14], while inhibition of Nox significantly increases the area of developing cerebellar internal granule layer[15]. By contrast, persistently elevated ROS levels in FoxO null mice result in impaired self-renewal and premature depletion of NSCs[8]. NSCs and neurons maintain a certain level of DNA double-strand breaks (DDSBs) for proper function[16,17], but it is unclear whether ROS impart DDSBs through specific signal pathways. The Ncf1 subunit has been shown to activate three of the Nox isoforms (Nox1, 2, and 3) found in the brain[10,11,18,19]. Thus, disrupting Ncf1 can

[1]Department of Anatomy and Cell Biology, Carver College of Medicine, University of Iowa, Iowa City, IA 52242, USA. ✉e-mail: john-engelhardt@uiowa.edu

inactivate multiple Nox isoforms. We sought to delineate the Nox-dependent determinants of NSC fate and the mechanism by which ROS influence neurogenesis. Our approach utilized *Ncf1* deletion for efficient inhibition of ROS production, and neurosphere formation as an index of lineage commitment toward dedicated neurosphere progenitors[20,21].

In this work, we show that Ncf1-dependent ROS increases the bioavailability of Igfbp2, presumably through oxidation of cysteine 43 in Igfbp2. Igfbp2 in turn acts in an autocrine/paracrine manner to repress DNA repair pathways required for NSC neurosphere formation and the specification of neurons. Genetic disruption of *Ncf1* or *Igfbp2* enhances the expression of DNA damage response genes and the self-renewal of NSCs which fail to form neurospheres and differentiate with a bias toward oligodendrocytes. Our study provides support for a direct link between Ncf1-dependent redox control of Igfbp2 signaling and the downstream regulation of NSC fate decisions through suppression of the Brca1 DNA damage responses.

## Results

### Lack of Ncf1 reduces ROS and promotes proliferation in mouse subventricular zone (SVZ)

We first evaluated whether Ncf1 controls ROS production in mouse brain. Indeed, ROS levels in the subventricular zone (SVZ) and rostral migratory stream (RMS) of hydroethidine-treated *Ncf1*[−/−] mice were significantly lower than WT mice (Supplementary Fig. 1a−c). To better understand the in vivo impact of Ncf1 on NSC biology, we evaluated proliferation in neurogenic niches of the adult mouse brain. Neurogenesis in the adult mammalian brain persists in two major niches, the subgranular zone (SGZ) of the dentate gyrus (DG) in the hippocampus and the SVZ of the lateral ventricle[22]. Adult *Ncf1*[−/−] mice incorporated significantly more BrdU in the SVZ and RMS than WT mice, and this significantly correlated with elevated levels of the neurotrophic factor cystatin C in the cerebrospinal fluid of *Ncf1*[−/−] mice (Supplementary Fig. 2a−c). Cystatin C is an extracellular cysteine protease inhibitor that stimulates the proliferation and self-renewal of NPCs[23], and supports neuroprotection[24].

These findings support a growing body of work implicating the redox state as a determinant of neural precursor cell fate[25]. Reducing conditions stimulate proliferation of NPCs, whereas mild oxidation has the opposite effect[26]. However, previous work has demonstrated that *Nox2*-deficient mice harbor reduced NPCs in the adult brain and produce fewer neurospheres in culture[27]. To better understand these differences between *Ncf1*[−/−] and *Nox2*[−/−] mice, we embarked on a series of in vitro studies evaluating NSC properties in culture.

### Ncf1 promotes NSC neurospheres formation and specification of neurons

We hypothesized that the lower levels of ROS in *Ncf1*[−/−] NSCs would alter their in vitro growth properties and behavior. When plated at clonal density, single NSCs undergo self-renewal and specify committed progenitors that form neurospheres[20,21]. In contrast to WT NSCs, which formed abundant neurospheres in culture, *Ncf1*[−/−] NSCs grew in sheets and formed fewer neurospheres (Fig. 1a, b). In addition, *Ncf1*[−/−] NSCs expanded more rapidly than WT NSCs in culture (Fig. 1c). This led us to hypothesize that deletion of *Ncf1* enhances proliferation of NSCs, while repressing specification toward less proliferative dedicated progenitors that comprise the neurosphere. To test this, we assessed EdU incorporation, cell death, and the differentiation profile of NSCs. *Ncf1*[−/−] NSCs showed higher proliferation and decreased cell death compared to WT NSCs (Fig. 1d−f). Differentiation of NSCs in culture produces neurons, astrocytes, and oligodendrocytes[28]. Interestingly, the granule cell layer (GCL) of the olfactory bulb of *Ncf1*[−/−] mice contained increased numbers of newly born (EdU+) oligodendrocytes and fewer EdU+ neurons than WT counterparts (Supplementary Fig. 3a−f). Similarly, differentiation of *Ncf1*[−/−] NSCs resulted in

increased numbers of oligodendrocytes with decreased neurons (Supplementary Fig. 3g−k). Thus, Ncf1 reduces proliferation of NSCs while promoting their commitment to neurogenesis.

We next investigated whether *Ncf1* functioned in a cell-autonomous manner to regulate lineage commitment of NSCs to form neurospheres. Coculture of transgene-marked *Ncf1*[−/−] NSCs with WT counterparts promoted the formation of clonal *Ncf1*[−/−] and chimeric neurospheres (Fig. 1g, h). When NSCs of the two genotypes were separated by a transwell membrane, neurosphere formation by the *Ncf1*[−/−] genotype was enhanced (Fig. 1g, h). Thus, Ncf1 promotes neurosphere formation in a non-cell-autonomous manner via a secreted factor.

### Ncf1-dependent ROS regulates the biologic activity of Igfbp2 and its ability to promote neurosphere formation from NSCs

To identify the Ncf1-dependent secreted factor required for neurosphere formation, we evaluated the NSC secretome by performing LC-MS/MS on WT and *Ncf1*[−/−] NSC cultures. We found that insulin-like growth factor-binding protein 2 (Igfbp2) was undetectable in the secretome of *Ncf1*[−/−] NSCs (Supplementary Data 1) and immunoblotting confirmed that Igfbp2 levels were significantly lower in *Ncf1*[−/−] conditioned media compared to WT counterparts (Fig. 1i). Reasoning that Ncf1-dependent ROS influence the function and/or secretion of Igfbp2 by NSCs, we treated *Ncf1*[−/−] NSCs with Igfbp2 or $H_2O_2$. Both conditions rescued neurosphere formation (Fig. 2a, b). The addition of Igfbp2 to WT NSCs resulted in the production of a greater number of neurospheres, and treatment with $H_2O_2$ had no effect on WT NSC cultures (Fig. 2b). Notably, regardless of genotype, treatment with $H_2O_2$ enhanced accumulation of Igfbp2 in the medium (Fig. 2b), suggesting that the production or secretion of Igfbp2 is redox-dependent. Moreover, supplementing the culture medium of *Ncf1*[−/−] NSCs with native or oxidized Igfbp2, but not reduced Igfbp2, induced neurosphere formation (Fig. 2c, d). In addition, *Igfbp2*[−/−] NSCs behaved similarly to *Ncf1*[−/−] NSCs; both genotypes grew as self-renewing sheets and formed very few neurospheres, and transfection of *Igfbp2*[−/−] NSCs with a plasmid harboring WT *Igfbp2* rescued neurospheres formation (Fig. 2e, f). To confirm that Igfbp2 is the downstream effector of ROS, we treated *Igfbp2*[−/−] NSCs with $H_2O_2$ and observed no restoration in neurosphere formation (Fig. 2g, h). In addition, the GCL of *Igfbp2*[−/−] olfactory bulb showed a phenotype similar to *Ncf1*[−/−] with more nascent oligodendrocytes and fewer emerging neurons than WT counterparts (Supplementary Fig. 3a−f). Together, these data indicate that Ncf1 regulates the commitment of NSCs to form neurospheres and neurons by modifying the redox-dependent secretion and biologic activity of Igfbp2.

### Ncf1/Igfbp2 signaling regulates the Brca1 DNA damage response pathway to control NSC lineage-commitment

To further characterize the Ncf1/Igfbp2-dependent phenotypic changes in early (48 h) NSC cultures, we compared the transcriptomes of WT and *Ncf1*[−/−] NSCs. We identified 4329 differentially regulated genes (Fig. 3a, b, Supplementary Data 2c) and found the gene ontogeny pathway *cell cycle regulation* to be the most significantly changed (Supplementary Data 3a). Other highly significant pathways included those involved in DNA damage responses: *cell cycle regulation of chromosomal replication*, *mitotic roles of polo-like kinase*, and *cell cycle: G2/M DNA damage checkpoint regulation* (Supplementary Fig. 4a−d and Supplementary Data 3a). Notably, the addition of Igfbp2 to *Ncf1*[−/−] NSC cultures shifted the expression of 566 genes to a level that was no longer significantly different from WT NSCs (Fig. 3c and Supplementary Data 2d), and these differences in expression correlated with the restoration of neurosphere formation. Pathway analysis of this subset of genes revealed the *role of Brca1 in DNA damage response* as the most changed pathway. Among these genes were members of the Fanconi anemia (FA) core complex (*Fanca*, *Fancc*, and *Fancl*), *Rad51*, *Mlh1*,

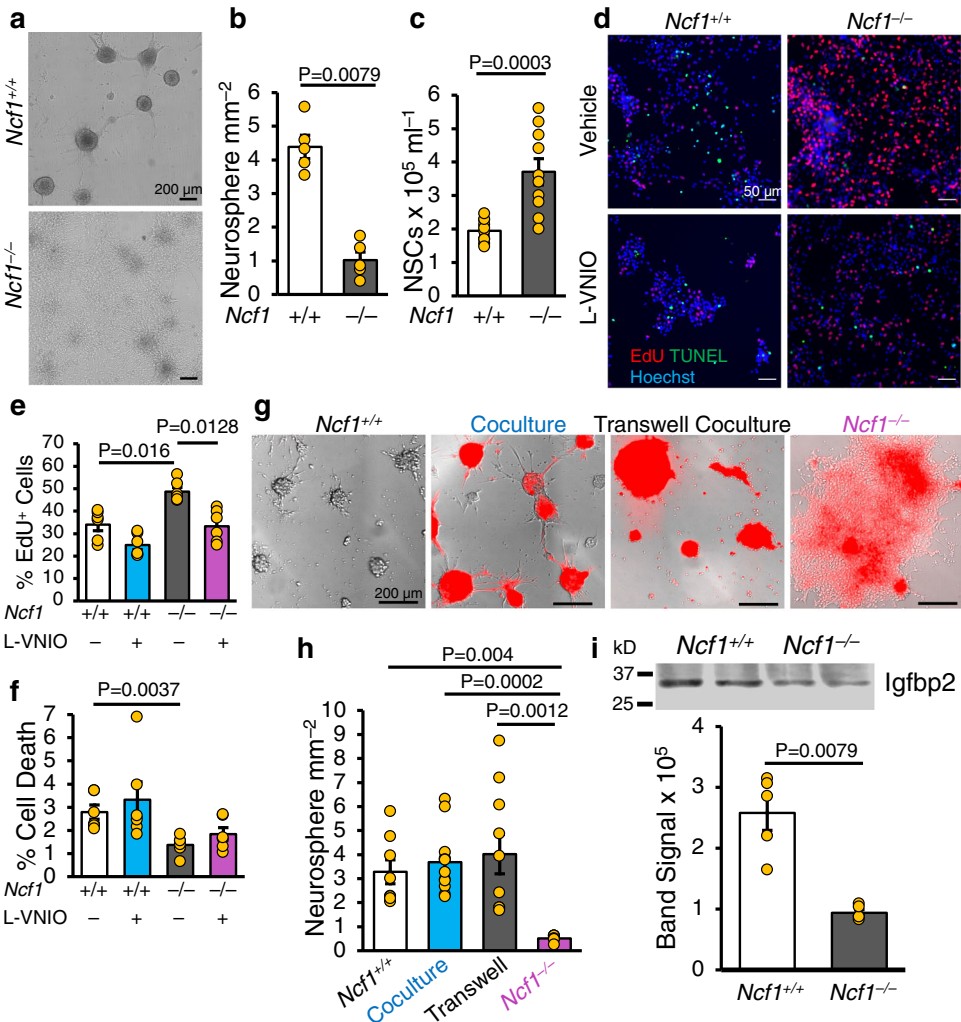

**Fig. 1 | The Nox2 subunit Ncf1 facilitates neurosphere formation in a non-cell autonomous manner by regulating Igfbp2 secretion and activity. a–c** Non-adherent cultures of primary postnatal forebrain NSCs were established from WT and $Ncf1^{-/-}$ mice and 2nd passage NSCs were grown for 11 days. **a** Photomicrographs of NSCs after 11 days in culture. **b** Quantification of neurospheres ≥60 μm in diameter ($n = 5$ donors). **c** Quantification of total cells in each NSC culture ($n = 10$ donors). **d–f** Monolayer cultures of 2nd passage neonatal NSCs were grown for 3 days with L-VNIO (100 μM) or vehicle. NSCs were pulsed with EdU (5 μM) for 4 h before fixation. **d** Photomicrographs of EdU (Red) and TUNEL (Green) labeling. **e** Quantification of EdU+ NSCs. **f** Quantification of cell death ($n = 6$ donors). **g** Photomicrographs of NSC cultures 5 days after plating 2nd passage of $Ncf1^{-/-}$

Tomato+ NSCs, in either direct coculture or transwell insert coculture with $Ncf1^{+/+}$ Tomato- NSCs. **h** Quantification of neurospheres ≥60 μm in diameter ($n = 8, 12, 10$, and 6 donors in WT, coculture, transwell coculture, and $Ncf1^{-/-}$ groups, respectively). **i** Western blot analysis and quantification comparing Igfbp2 levels in the conditioned medium 5 days after plating ($n = 5$ donors). The samples were derived from the same experiment and blots were processed in parallel. kD = kilodalton. Two-tailed Mann–Whitney U test (**b, c, i**) and Kruskal–Wallis (see Source data for full details) followed by Benjamini–Hochberg FDR multiple comparison posttest (**e, f, h**). Error bars indicate s.e.m. Scale: 200 μm. Source data are provided as a Source data file.

*Atrip*, and *Rfc2*; all were expressed at significantly higher levels in $Ncf1^{-/-}$ vs. WT NSCs. Upon treatment with Igfbp2, these genes were downregulated and returned to WT levels (Fig. 3d, Supplementary Fig. 5). Moreover, knockdown of *Fanca*, *Fancd2*, or *Rad51* in $Ncf1^{-/-}$ NSCs (Supplementary Fig. 7a–c) was sufficient to restore neurosphere formation (Fig. 4a, b). This observation was also confirmed in adult $Ncf1^{-/-}$ and $Igfbp2^{-/-}$ NSCs. Knocking down *Fanca*, *Fancd2*, or *Rad51* in adult $Ncf1^{-/-}$ and $Igfbp2^{-/-}$ NSCs rescued neurosphere formation to WT levels. However, knocking down these same genes in WT NSCs did not alter the frequency of neurosphere formation (Supplementary Fig. 8a–c).

NSC self-renewal and maintenance of a multipotent stem cell state is classically defined by the ability of a single NSC to repeatedly form neurospheres on serial passages. We hypothesized that Ncf1/Igfbp2/Brca1 axis controls the ability of NSCs to exit a self-renewing state to differentiate into dedicated NPCs required to form neurospheres. To this end, we investigated whether the sheets formed by $Ncf1^{-/-}$ and

$Igfbp2^{-/-}$ NSCs were capable of self-renewal as defined by neurosphere formation potential on serial passage. Since $Ncf1^{-/-}$ and $Igfbp2^{-/-}$ NSCs do not spontaneously form neurosphere, we inhibited Rad51 in these cultures to induce neurosphere formation prior to each serial passage. Recurrent treatment of $Ncf1^{-/-}$ and $Igfbp2^{-/-}$ NSCs with Rad51 dsiRNA was sufficient to restore neurosphere formation to WT levels for three successive passages (Supplementary Fig. 8d–h). Thus, we conclude that disruption of $Ncf1$ and $Igfbp2$ enhances self-renewal of NSCs while repressing lineage-commitment toward dedicated progenitors.

These findings suggested that Ncf1-dependent redox modification of Igfbp2 directly regulates DNA damage response pathways in NSCs and that inhibition of DNA repair is the terminal Igfbp2-signaling event required for neurosphere formation. To further support this mechanism, we treated $Ncf1^{-/-}$ and $Igfbp2^{-/-}$ NSCs with $H_2O_2$ or Igfbp2 and evaluated changes in *Fanca*, *Fancd2* and *Rad51* expression. Whereas Igfbp2 treatment suppressed expression of these three genes in both $Ncf1^{-/-}$ and $Igfbp2^{-/-}$ NSCs, $H_2O_2$ treatment repressed *Fanca*,

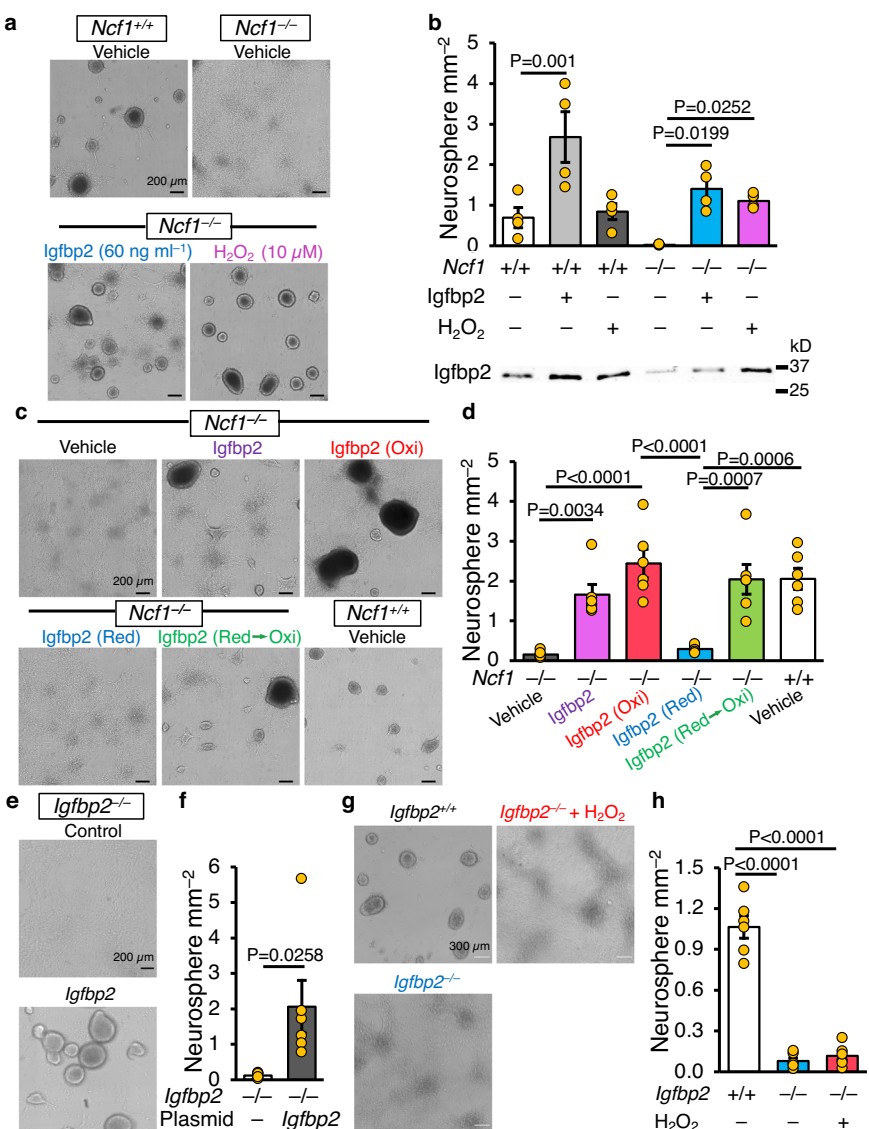

**Fig. 2 | Non-reduced Igfbp2 is necessary for NSC lineage commitment.**
**a**, **b** Second-passage WT and *Ncf1*⁻/⁻ NSCs were treated with mouse recombinant insulin-like growth factor-binding protein 2 (Igfbp2) or H₂O₂ 24–28 h after plating. **a** Photomicrographs of NSCs 5 days after treatment with Igfbp2 (60 ng ml⁻¹) or H₂O₂ (10 μM). **b** Quantification of neurospheres ≥60 μm in diameter (*n* = 4 donors). Western blot below the bar graph shows Igfbp2 levels in the medium 5 days post-treatment. **c** Photomicrographs of NSCs 8 days after treatment with 100 ng ml⁻¹ native Igfbp2, H₂O₂-treated (Oxi), DTT-treated (Red), or DTT then H₂O₂ sequentially treated (Red → Oxi) Igfbp2. **d** Quantification of neurospheres ≥60 μm in diameter (*n* = 6 donors). **e**, **f** *Igfbp2*⁻/⁻ NSCs were transfected with empty or Igfbp2 expressing plasmid 24 h after plating. **e** Photomicrographs of NSCs 11 days after transfection. **f** Quantification of neurospheres ≥60 μm in diameter (*n* = 6 donors). **g**, **h** Second passage WT and *Igfbp2*⁻/⁻ NSCs were treated with H₂O₂ 24 h after plating. **g** Photomicrographs of NSCs 10 days after H₂O₂ (10 μM) treatment. **h** Quantification of neurospheres ≥60 μm in diameter (*n* = 6 donors). Two-tailed students t-test (**f**), Two-way ANOVA (**b**), and One-way ANOVA (**d**, **h**), followed by Bonferroni multiple comparison posttest. Error bars indicate s.e.m. Scale: 200 μm (**a**, **c**, **e**) and 300 μm (**g**). Source data are provided as a Source data file.

*Fancd2*, and *Rad51* expression only in *Ncf1*⁻/⁻ NSCs, demonstrating that Igfbp2 was required for the redox-dependent suppression of these DNA repair genes (Supplementary Fig. 7d–i).

The activity of neuronal nitric oxide synthase (nNos), which is expressed in neurons and NSCs, supports neurogenesis[29]. Superoxide anions react with nNos-generated nitric oxide (NO) to produce the potent oxidant peroxynitrite (ONOO⁻) that nitrosylates thiol groups (S-Nitrosylation)[30], emphasizing the potential importance of nNos and Nox in coordinating NSC behavior. To evaluate potential links between nNos and Nox that might explain our current findings, we performed biotin derivatization to detect S-Nitrosylation in the SVZ of WT, *Ncf1*⁻/⁻, and *Igfbp2*⁻/⁻ mice. The SVZ of *Ncf1*⁻/⁻ had lower levels of S-Nitrosylation than WT and *Igfbp2*⁻/⁻ counterparts (Supplementary Fig. 6). Thus, decreased S-Nitrosylation in *Ncf1*⁻/⁻ SVZ may be due to a reduction in

superoxide bioavailability. Inhibition of nNos, using N5-(1-Imino-3-butenyl)-L-ornithine (L-VNIO), inhibited proliferation of *Ncf1*⁻/⁻ NSCs toward that observed in WT counterparts, while there was no effect on cell death (Fig. 1d–f). However, it remains unclear if nNos inhibition of proliferation was specific to *Ncf1*⁻/⁻ NSCs, given the drug also generally reduced cell numbers in WT NSCs. Notably, L-VNIO treatment had no effect on *Fanca*, *Fancd2*, and *Rad51* expression in either *Ncf1*⁻/⁻ or *Igfbp2*⁻/⁻ NSCs (Supplementary Fig. 7d–i). Given that nNos inhibition reduced proliferation of *Ncf1*⁻/⁻ NSCs (Fig. 1d–f), but did not alter the expression of these DNA repair genes, we believe that the Ncf1/ROS/Igfbp2/DNA repair axis regulates NSCs proliferation independently of nNos-generated NO. In support for this finding, in vivo S-Nitrosocysteine abundance in the SVZ of *Igfpb2*⁻/⁻ was similar to WT mice (Supplementary Fig. 6), despite *Igfpb2*⁻/⁻ phenocopying the

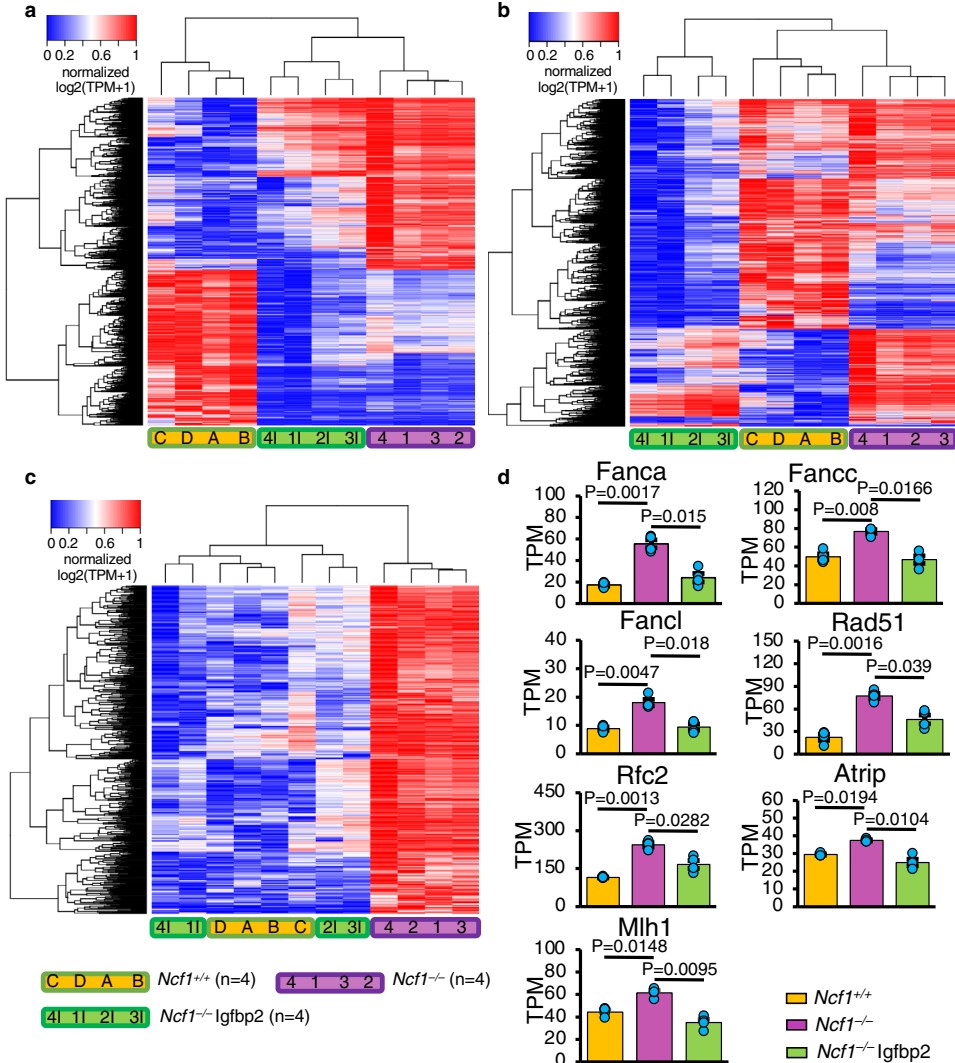

**Fig. 3 | Ncf1 represses NSC DNA repair pathways in an Igfbp2-dependent manner.** NSCs prepared from 4 WT (A, B, C, and D) and 4 *Ncf1*−/− mice were treated with vehicle (1, 2, 3, and 4) or Igfbp2 (1I, 2I, 3I, and 4I) 48 h after plating. Vehicle- and Igfbp2-treated *Ncf1*−/− groups are matched for NSC preparation by the number with "I" indicating Igfbp2-treatment. After additional 48 h, total RNA was collected from all groups and ribosome-depleted RNAseq was performed. **a–c** Heat maps represent clustering based on Euclidean distance of all genes that were significantly differentially expressed between any of the three groups (WT, *Ncf1*−/−, and Igfbp2-treated *Ncf1*−/− NSCs) (**a**), WT and *Ncf1*−/− (**b**), and WT and *Ncf1*−/− for which expression was restored towards WT levels after treatment with Igfbp2 (**c**). Statistical analysis

of changes in gene expression used Benjamini−Hochberg FDR corrected one-way ANOVAs (**a**) and Tukey's post hoc tests (**b**, **c**). **d** Ingenuity Pathway Analysis (IPA) performed on the gene sets in (**c**) using the absolute fold change for *Ncf1*−/− vs. WT and *Ncf1*−/− + Igfbp2 vs. *Ncf1*−/− revealed the top pathway: Role of Brca1 in DNA Damage Response (P = 0.0010 and P = 0.0017, respectively) (Supplementary Data 3b). Subset of genes in the Brca1 pathway that are expressed differentially in each of the experimental groups. TPM: transcript per million. Benjamini−Hochberg corrected One-way ANOVA followed by Tukey's multiple comparison post hoc test, N = 4 donors. Source data are provided as a Source data file.

altered specification of oligodendrocytes and neurons in the olfactory bulb of *Ncf1*−/− mice (Supplementary Fig. 3a–f). We attempted to investigate the effect of nNos inhibition on neurosphere formation in WT, *Ncf1*−/−, and *Igfbp2*−/− NSCs; however, NSCs of all genotypes did not tolerate prolonged exposure to L-VNIO, suggesting a reduction in proliferation could be an off-target consequence of toxicity.

We next sought to determine whether Ncf1/Igfbp2-dependent inhibition of the DNA repair pathways was linked to DNA damage in NSCs. γ-H2ax is a variant of Histone H2A that binds to DNA double-strand breaks (DDSBs)[31]. Notably, *Ncf1*−/− NSCs maintained low levels of γ-H2ax and supplementation with Igfbp2 elevated the levels of γ-H2ax toward that found in WT NSCs (Fig. 4c, d). Similarly, Nestin+ NSCs in the SVZ of adult *Ncf1*−/− and *Igfbp2*−/− mice had higher expression of Rad51 and lower γ-H2ax than WT counterparts (Supplementary Fig. 9a–c). In addition, immunoblotting of NSC lysates showed that *Ncf1*−/− and *Igfbp2*−/− NSCs express higher levels of Nestin and Rad51 proteins but

lower levels of γ-H2ax than WT counterparts (Supplementary Fig. 9d–h). Furthermore, dsiRNA-mediated knockdown of *Fanca*, *Fancd2*, or *Rad51* in *Ncf1*−/− NSCs increased γ-H2ax levels (Fig. 4e, f) and neurosphere formation (Fig. 4a, b) to levels observed in WT NSCs and decreased proliferation and increased cell death of *Ncf1*−/− NSCs back to WT levels (Supplementary Fig. 9i–k). In addition, overexpression of *Fanca* in WT NSCs significantly decreased neurosphere formation (Fig. 4g–i). These findings link Ncf1/Igfbp2-mediated modulation of DNA repair in NSCs to their ability to form committed NPCs−containing neurospheres.

Differentiated *Ncf1*−/− NSCs gave rise to a lower percentage of neurons and a higher percentage of oligodendrocytes than their WT counterparts (Supplementary Fig. 3g–k), suggesting that persistent inhibition of ROS production promotes oligodendrogenesis. These changes in the differentiation profile of *Ncf1*−/− and WT NSCs were mirrored in the analysis of differentially expressed genes. For example,

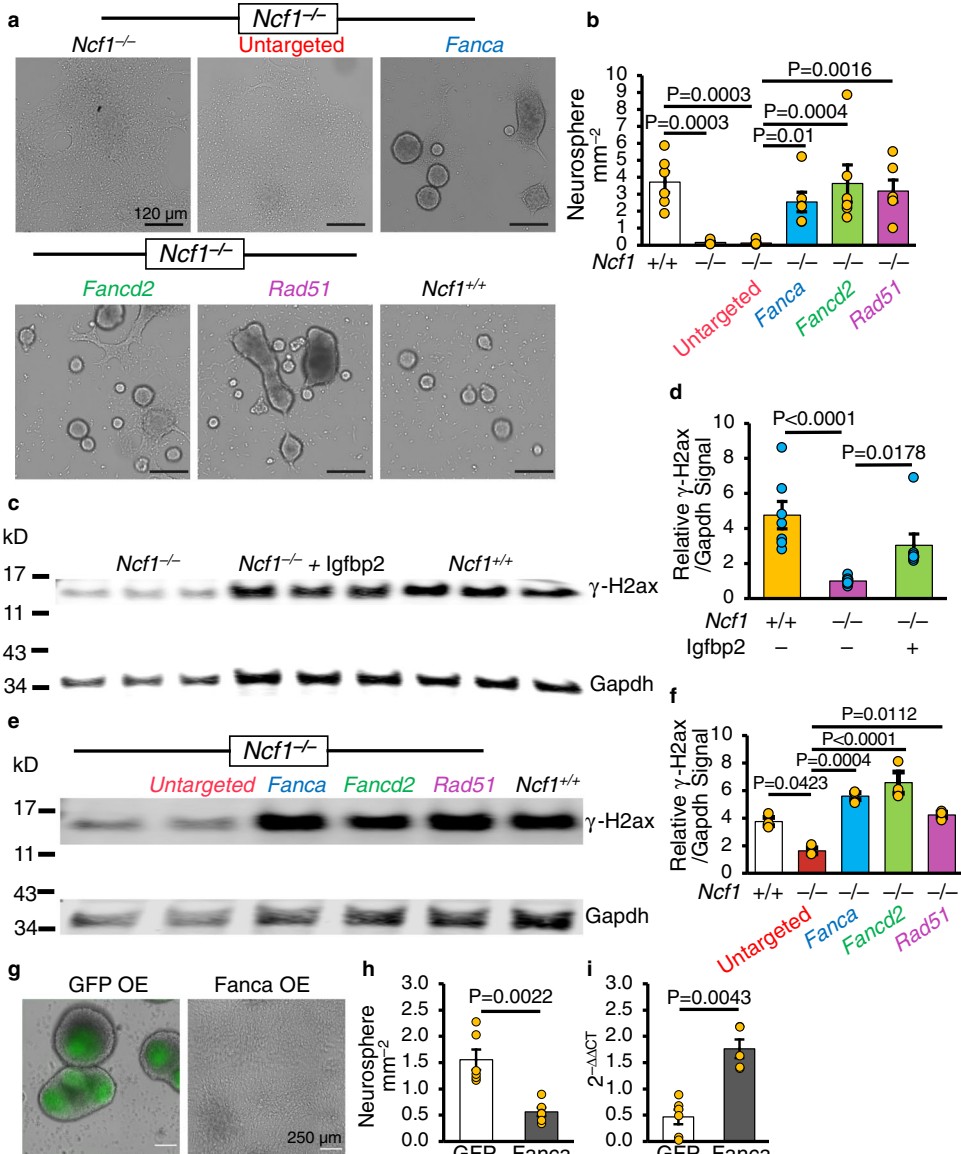

**Fig. 4 | Ncf1/Igfbp2 axis promotes NSC lineage-commitment through repression of NSC DNA repair pathways. a, b** Photomicrographs (**a**) and quantification (**b**) of neurospheres ≥60 μm in diameter 7 days after transfection of *Ncf1⁻/⁻* NSCs with siRNA targeting Fanca, Fancd2 or Rad51 or negative control (*n* = 6 donors). **c, d** Western blot analysis (**c**) and quantification (**d**) of γ-H2ax levels (normalized to Gapdh) in NSC lysates 3 days after Igfbp2 treatment (*n* = 7 donors from two different experiments). **e, f** Western blot (**e**) and quantification (**f**) of γ-H2ax levels (relative to Gapdh) in cell lysates 3 days after transfection (*n* = 3 donors). The samples were derived from the same experiment and blots were processed in parallel. kD = kilodalton. **g**–**i** WT NSCs were transduced with Peggy Back transposase and transposon to overexpress GFP or Fanca. **g** Photomicrographs and

**h** quantification of neurospheres ≥60 μm in diameter ten days after plating P4 NSCs overexpressing GFP or Fanca (*n* = 6 donors). **i** qPCR of *Fanca* mRNA 3 days after plating P5 WT NSCs overexpressing GFP or Fanca (*n* = 6 and 5 donors in GFP and Fanca OE NSCs respectively). ΔΔCT: the difference in threshold cycles normalized to β-Actin. Kruskal–Wallis test and FDR method of Benjamini and Hochberg multiple comparison posttest (**d**), One-way ANOVA and FDR method of Benjamini and Hochberg (**b**) or Bonferroni multiple comparison posttest for marked comparisons in (**f**) and two-tailed Mann–Whitney U test (**h**, **i**). Error bars indicate s.e.m. Scale: 120 and 250 μm in (**a**) and (**g**) respectively. Source data are provided as a Source data file.

several nervous system development pathways, including the *development of oligodendrocytes* and *development of neurons* were differentially regulated (*P* < 0.001) (Supplementary Fig. 4e–h and Supplementary Data 3c). Knockdown of *Fanca*, *Fancd2*, or *Rad51* in *Ncf1⁻/⁻* NSCs enhanced neurogenesis and inhibited oligodendrogenesis (Supplementary Fig. 3l–m), further supporting the notion that Ncf1/Igfbp2-mediated repression of the DNA damage response promotes the commitment of NSCs toward neuronal fates.

**Igfbp2 cysteine 43 is required for NSC lineage-commitment**

Although the secretome of *Ncf1⁻/⁻* NSCs contained lower levels of Igfbp2 (Fig. 1i), their transcriptome showed higher levels of *Igfbp2*

mRNA compared to WT NSCs (6.7-fold, *P* < 0.0012) (Supplementary Data 2a). In addition, treating *Ncf1⁻/⁻* NSCs with H₂O₂ promoted Igfbp2 secretion and neurosphere formation (Fig. 2a, b). This led us to hypothesize that Ncf1-dependent ROS post-transcriptionally regulate Igfbp2 function. Labeling of oxidized, native, or reduced recombinant mouse Igfbp2 (rmIgfbp2) with biotinylated iodoacetamide (BIAM) resulted in greater biotinylation of the reduced (TCEP-treated) compared to native Igfbp2 and both showed higher biotinylation than the oxidized (H₂O₂-treated) Igfbp2 (Fig. 5a, b), suggesting the presence of redox-sensitive cysteines. Given that native and oxidized, but not reduced, Igfbp2 rescues neurosphere formation by *Ncf1⁻/⁻* NSCs (Fig. 2c, d), we hypothesized that

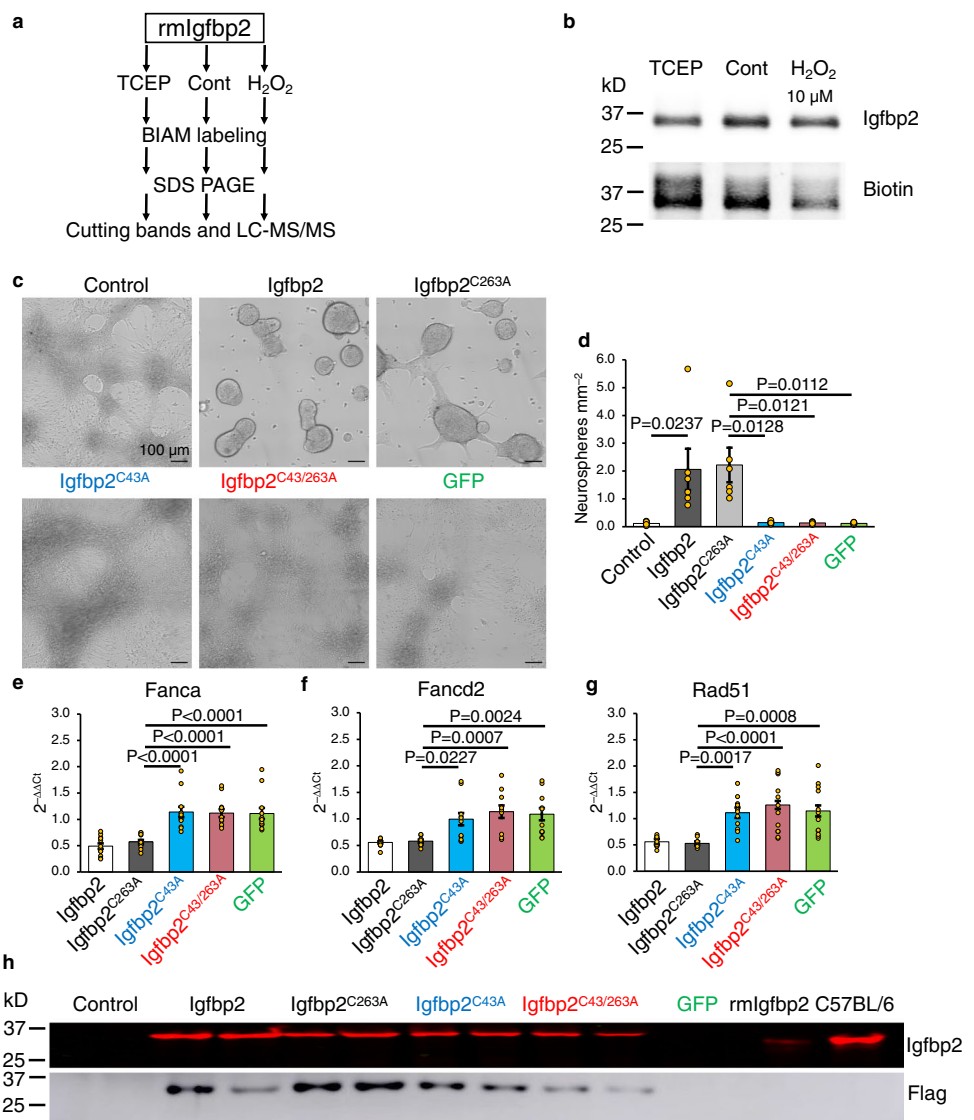

**Fig. 5 | Ncf1/Igfbp2-mediated repression of NSC DNA repair pathways requires Igfbp2^C43 to promote neurosphere formation. a** Experimental design. **b** Western blot analysis of untreated or TCEP- or $H_2O_2$-treated recombinant mouse Igfbp2 (rmIgfbp2) after labeling with biotinylated iodoacetamide (BIAM). **c–h** *Igfbp2^{-/-}* NSCs were transfected with Igfbp2, Igfbp2^C263A, Igfbp2^C43A, Igfbp2^C43/263A, or GFP expressing plasmid 24 h after plating. **c** Photomicrographs of NSCs 11 days after transfection. **d** Quantification of neurospheres ≥60 μm in diameter (*n* = 6 donors). **e–g** qPCR of Fanca (**e**), Fancd2 (**f**) and Rad51 (**g**) in *Igfbp2^{-/-}* NSCs 5 days after transfection with Igfbp2, Igfbp2^C263A, Igfbp2^C43A, Igfbp2^C43&263A, or GFP expressing plasmid (*n* = 12 donors). ΔΔCT: the difference in threshold cycles normalized to β-Actin and compared to untransfected *Igfbp2^{-/-}* NSCs. **h** Western blot analysis of Igfbp2 and flag-tag in conditioned medium 3 days after transfection. kD = kilodalton. One-way ANOVA followed by Bonferroni multiple comparison posttest for marked comparisons in (**d**) or comparing the mean of each group to the mean of untransfected *Igfbp2^{-/-}* NSCs in (**e–g**). Error bars indicate s.e.m. Scale: 100 μm. Source data are provided as a Source data file.

oxidation of cysteine residue(s) in Igfbp2 might control its biologic activity. LC-MS/MS analysis of these BIAM-labeled Igfbp2 substrates identified two cysteines, C43 and C263, as carrying reduced thiol groups only under reducing conditions (Supplementary Fig. 10a, b). Expression of WT Igfbp2 and mutant Igfbp2^C263A, but not Igfbp2^C43A or Igfbp2^C43A/C263A, in *Igfbp2^{-/-}* NSCs restored neurosphere formation (Fig. 5c, d) and led to the downregulation of *Fanca*, *Fancd2*, and *Rad51* (Fig. 5e–g). Western blot analysis confirmed the expression and secretion of WT and mutant Igfbp2 in the conditioned medium of *Igfbp2^{-/-}* NSCs (Fig. 5h). Together, these data show that C43 is required for neurosphere formation and suggest that oxidation of C43 in Igfbp2 is essential for downregulating the NSC DNA damage response required for lineage commitment, neurosphere formation, and neurogenesis.

Igf1 has been demonstrated to impact NSC fate decisions by binding Igf1R and subsequently modulating the PI3K/Akt pathway[32–34]. Thus, we hypothesized that the redox state of Igfbp2 might regulate NSC growth and fate by altering its binding affinity to Igf1 and/or Igf2. We evaluated the binding affinity of native, reduced (DTT-pretreated), or oxidized ($H_2O_2$-pretreated) Igfbp2 to Igf1 and Igf2 (Supplementary Fig. 10c, d) and observed no redox-dependent changes in Igfbp2 binding to Igf1 or Igf2. These results support an Igf-independent mechanism for Igfbp2 regulation of NSC growth and cell fate.

## Discussion

Our findings demonstrate that Ncf1-dependent oxidation of Igfbp2 represses DNA repair pathways in self-renewing NSCs, and that this event is required for their commitment toward neuronal precursors. Given the importance of DNA repair in NSC maintenance, precise regulation of Ncf1-dependent ROS is likely central to orchestrating the balance between self-renewal and lineage commitment. For example,

persistent reducing conditions have been previously shown to promote NSC proliferation, while the opposite effect was observed in oxidizing conditions[26]. These findings are consistent with our present study demonstrating that Igfbp2-C43 was required for the repression of DNA repair genes (Fanca, Fancd2, Rad51) in NSCs and this repression promoted the formation of DDSBs and exit from a self-renewing state. By contrast, our findings would predict that persistent ROS-dependent inhibition of DNA repair pathways would negatively impact NSC self-renewal and may explain why persistent elevation of ROS leads to NSC senescence[1,8]. However, it remains unclear whether DDSB induction is a requirement for lineage commitment or simply a consequence of the repression of cell cycle and DNA repair checkpoints that trigger the commitment process. Recent findings have suggested that DDSBs control the expression of neuronal genes required for learning and memory[31], and that NSCs maintain frequent DNA break clusters at long transcribed and late-replicating neural genes[17]. Such directed DNA damage/repair pathways may have overlapping biology with the ability of Ncf1/Igfbp2 to control DNA damage, self-renewal, lineage commitment, and differentiation of NSCs in a redox-dependent manner.

We propose that the repression of Ncf1/Igfbp2 activity in NSCs governs a self-renewal program in which DNA repair pathway activity is protective, and that exit from this self-renewing state and commitment toward neuronal lineages requires activation of Nox and Igfbp2-mediated transcriptional repression of the Brca1 DNA damage response pathway. Brca1 is essential for brain development because it inhibits NSC apoptosis[35], and Fanca is required for the survival and maintenance of these cells[36]. The Brca1 pathway is also linked to cell cycle checkpoints[37] and chromatin remodeling[38], consistent with the influence of Ncf1/Igfbp2 on NSC proliferation and lineage commitment. Notably, direct inhibition of Brca1 DNA repair genes was sufficient to restore normal NSC behavior to Ncf1$^{-/-}$ and Igfbp2$^{-/-}$ NSCs, and overexpression of Fanca resulted in decreased neurosphere formation by WT NSCs while promoting a self-renewing state, confirming that DNA repair pathways and/or DDSBs are the distal most events orchestrating Ncf1/Igfbp2-dependent instructions to NSC commitment decisions.

Despite the well-recognized importance of ROS in NSC biology, the effects of Nox2-dependent ROS on the maintenance and cell-fate decision of NSCs are widely debated. Some reports suggest that FGF2 stimulates self-renewal and proliferation of NSCs through induction of Nox2, and that high levels of Nox2-dependent ROS are essential for self-renewal and neurogenesis through oxidation of PTEN and activation of AKT[6,9]. Igfbp2 can regulate cell growth by binding and sequestering Igf1/2, but its biological activities can also be independent of Igfs[39]. In our studies, the binding affinity of Igf1/2 to Igfbp2 was unaffected by its redox state, suggesting that Igfbp2 regulation of NSCs is independent of Igfs. Conversely, other reports indicate that FOXO3 maintains NSC quiescence and prevents premature senescence by preventing high ROS levels[8]. In addition, lowering ROS levels have been suggested to improve self-renewal and proliferation of NPCs through downregulation of Sirt1[26].

ROS have been shown to regulate NSC growth, lineage-commitment, differentiation and senescence through multiple pathways[6,8,9,26]. However, our studies show that Igfbp2-deficient NSCs behave similar to Ncf1 null NSCs with higher proliferation, lower lineage commitment, elevated DNA repair pathways, and reduced DNA damage than WT NSCs. Whereas $H_2O_2$-treatment restored WT levels of NSC proliferation, enhanced cell death, and promoted lineage commitment in the absence of Ncf1, $H_2O_2$-treatment had no effect on Igfbp2 null NSCs, demonstrating that Igfbp2 was required for the ROS-dependent changes in NSC behavior. Moreover, directly supplementing Ncf1 deficient NSCs with oxidized, but not reduced, Igfbp2 or transient knockdown of DNA repair genes restored WT levels of NSC proliferation, death, DDSBs, and lineage commitment. Furthermore,

mutagenesis of Igfbp2 demonstrated that cysteine 43 was required for normal NSC behavior. These findings highlight the importance of Igfbp2 and DNA repair networks as downstream effectors for Ncf1-dependent ROS signaling.

Overall, our study provides insights into how Ncf1-dependent changes in the cellular redox state coordinate cell cycle progression and fate decisions by NSCs through Igfbp2. Whether ROS-mediated DNA damage and repair pathway also control cell-cycle progression and lineage commitment in other TSSCs remains to be determined.

## Methods

### Animals
Unless otherwise indicated, 8–12-week-old C57BL/6J mice (The Jackson Laboratory), Igfbp2$^{-/-}$ mice (a kind gift from Dr. Clifford J. Rosen), and Ncf11$^{-/-}$ mice (Taconic) were used in this study. Ncf1$^{-/-}$ mice were back crossed to C57BL/6J mice for at least 10 generations, in house, before using in experiments. For neurosphere experiments, pups lacking Ncf1 and carrying the Gt(ROSA)26$^{tdTomato}$ transgene were generated by crossing B6.129(Cg)-Gt(ROSA)26Sor$^{tm4(ACTB-tdTomato,-EGFP)Luo}$/J mice (The Jackson Laboratory) to Ncf1$^{-/-}$ mice, and then mating the heterozygote offspring. Igfbp2$^{-/-}$ mice[40] were a kind gift from Dr. Clifford J. Rosen. Animals were typically group housed and maintained in 12-h light/dark cycle with set points at 73 degrees Fahrenheit and 50% humidity. Animals were randomly assigned to the control or treated groups throughout the study. Breeding and all mouse experiments were performed according to a protocol approved by the University of Iowa Institutional Animal Care and Use Committee (IACUC).

### Primary cell cultures
Primary neurosphere cultures were established from postnatal forebrain as described[41], with the following modifications. Mouse pups were decapitated at 1–2 days of age. Forebrains were dissected and triturated to a single-cell suspension with a glass pipette. Primary cells from each pup were cultured and propagated in serum-free medium (SFM) DMEM/F12 supplemented with EGF, FGF (Sigma), insulin, transferrin, sodium selenite (ITSS) (Roche) and B-27 supplement (Invitrogen). After neurospheres were established, they were collected, incubated in TrypLE (Invitrogen) for 15 min at 37 °C then triturated to a single-cell suspension using glass pipettes. Adult SVZ NSCs were prepared as described[42], with the following modifications. 8–12-week-old mice were euthanized using a high dose isoflurane followed by cervical dislocation. The SVZ was dissected and triturated to a single-cell suspension with a glass pipette. Cells were strained through a 70 μm cell strainer. Primary cells from each mouse were cultured and propagated in the same SFM used with neonatal NSCs. Second- or third-passage cells were used in all experiments. NSCs from each pup or adult were grown separately and the number of replicates in all cell culture experiments represents the number of pups or adult animals used.

### BrdU labeling of animals
Ten- to twelve-week-old mice received a single intraperitoneal injection of 5'-bromo 2'-deoxyuridine (BrdU 300 mg kg$^{-1}$). They were sacrificed 24 h later, and the brains were dissected and immediately imbedded in optimal cutting temperature compound (OCT) (Tissue-Tek). Serial fresh-frozen sagittal sections (20 μm) were cut using a cryostat. Sections were kept at −80 °C until used.

### Collection and analysis of cerebrospinal fluid (CSF)
CSF was collected as described[43] and kept at −80 °C until used. A mouse Cystatin-C ELISA kit (Biovendor) was used according to the manufacturer's instructions, with the following changes: 1 μl of each CSF sample was diluted 1:3000 in two steps (1:15 then 1:200) with the dilution solution provided in the kit; and the samples were incubated

with primary antibody overnight at 4 °C. A standard curve was generated and used to calculate the Cystatin-C concentration of the CSF.

## Measurement of endogenous reactive oxygen species (ROS) levels

In vivo ROS levels were measured using the ROS-sensitive dye hydroethidine (Invitrogen) as described[44]. Briefly, 12-week-old C57BL/6 or *Ncf1*[−/−] males received an intravenous injection of hydroethidine (10 mg kg[−1], in PBS 1% DMSO). Five hours later, the mice underwent intracardiac perfusion with PBS followed by 4% paraformaldehyde in PBS. Parasagittal sections of 30 μm thickness were cut using a cryostat. Nuclei were counterstained with Hoechst and coverslips were applied. Slides from different groups were imaged immediately, in parallel. Quantification was performed by Metamorph software and at the same threshold.

## Detection of cysteine S-Nitrosylation

In vivo cysteine S-Nitrosylation levels were measured using biotin derivatization as previously described[45]. Briefly, 20 μm parasagittal brain sections (every 20th) of 12-week-old C57BL/6, *Ncf1*[−/−], or *Igfbp2*[−/−] mice were fixed in 4% paraformaldehyde in PBS and washed three times with PBS containing 0.4 mM EDTA and 40 μM neocuproine. Free thiol groups were then blocked with 40 mM N-ethylmaleimide (NEM) in PBS containing 0.4 mM EDTA, 0.04 mM neocuproine, and 2.5% SDS for 30 min. Sections were washed three times and then incubated with 1 mM sodium ascorbate in PBS for 15 min to reduce S-nitrosylated proteins. Newly reduced cysteine residues were then labeled with 0.1 mM N-(3-Maleimidopropionyl)biocytin (MPB) in PBS for 30 min. After washing, sections were incubated with DyLight 549-conjugated streptavidin (1:250, Jackson Immunoresearch Labs) for 30 min. Nuclei were counterstained with Hoechst 33342 and coverslips were mounted in Aquamount. Slides from different groups were stained and imaged, in parallel. Quantification was performed by Metamorph software at the same threshold for all images.

## Immunofluorescence (IF)

Slides representing every twentieth section (at least 3 sections from each hemisphere), matching the parasagittal sections from all mice, were fixed in 4% PFA for 30 min at room temperature, washed 3 times in 1 x PBS, and boiled in 0.01 M sodium citrate buffer pH 6.0, for 25 min at 95–98 °C, for epitope retrieval. Sections were permeabilized in blocking solution (1 x PBS 20% donkey serum, 1 mM CaCl$_2$, and 0.5% Triton-X 100). The slides were then incubated with primary antibody against BrdU (mouse, 1:250, BD Bioscience) diluted in diluent solution (1 x PBS containing 0.5% Triton-X 100, 1% donkey serum and 1 mM CaCl$_2$, overnight at 4 °C). Subsequently, slides were washed 3 times for 10 min in 1 x PBS, and then incubated with Alexa Fluor 568-conjugated donkey anti-mouse IgG secondary antibody (Jackson Immunoresearch, 1:250) and diluted in the same solution for 2 h at room temperature. Slides were washed in 1 x PBS 3 times, for 10 min each. SlowFade Gold with DAPI (Invitrogen) was added and coverslips were applied. Immunofluorescence staining for γ-H2ax, Rad51, Nestin, NeuN, and O4 was done similarly with the exception that epitope retrieval was performed in a pressure cocker for 5 min for γ-H2ax, Rad51, and Nestin. Primary antibodies used were γ-H2ax (rabbit, 1:300, Abcam), Rad51 (mouse, 1:330, Abcam), and Nestin (chicken, 1:250, Novus Biologicals). Immunofluorescent NeuN and O4 staining was done without epitope retrieval using primary antibodies to NeuN (rabbit, 1:400, Abcam) and O4 (mouse, 1:440, Millipore).

## Neurosphere formation and growth

Second-passage NSCs from each pup were seeded at clonal density (100 cells ml[−1] of medium)[9] in uncoated 24-well plates (Nunc). The cultures were monitored daily for neurosphere formation. After 11 days in culture in SFM, differential interface contrast (DIC) or DIC

and red fluorescent protein (tdTomato) tiled scans of every well were acquired using the MetaMorph software (Molecular Devices) and a spinning-disk microscope DMI-60000 (Leica) equipped with an EMCCD camera (Hamamatsu). Cells were then harvested, triturated to a single-cell suspension, and counted using Countess (Invitrogen) according to the manufacturer's protocol.

## Immunofluorescence-based staining of neurospheres

Second-passage NSCs were seeded at clonal density (100 cells ml[−1] of media) in glass-bottom 35 mm dishes (MatTek) coated with poly-d-lysine and laminin. Five days after plating, developing neurospheres remained attached to the plate. At this time, they were grown in differentiation medium (DMEM/F12 supplemented with fetal bovine serum (FBS) and IGF (Sigma)). After 11 days in differentiation medium, the differentiated progenitors were fixed in 4% PFA for 1 h at room temperature. Differentiated progenitors were then carefully washed 3 times in 1 x PBS, using a bulb pipette with a 200 μl pipette tip attached to its end to avoid detachment. Differentiated progenitors were then incubated first in blocking solution for 1 h at room temperature, and then with primary antibodies against the following antigens: β-Tubulin III (rabbit, 1:250, Sigma), O4 (mouse, 1:440, Millipore) and Gfap (chicken, 1:250, Aves Labs) in diluent solution, overnight at 4 °C. Excess primary antibody was washed off 3 times using 1 x PBS. Differentiated progenitors were then incubated with fluorophore-conjugated secondary antibodies in diluent solution for 2 h at room temperature. Excess secondary antibody was then washed off using 1x PBS (three washes of 10 min duration each). SlowFade Gold with DAPI (Invitrogen) was then applied.

## Assessment of NSC proliferation and cell death

NSCs were seeded at density of 200,000 cells/2 ml onto poly-D-lysine coated 35 mm dishes with glass-bottom coverslips. NSCs were grown under proliferation conditions with Vinyl-L-NIO (L-VNIO, 100 μM) (Enzo Lifesciences) or vehicle for 3 days. 5-ethynyl-2′-deoxyuridine (EdU, 5 μM) was added to all dishes for 4 h before fixation in 4% paraformaldehyde. Cells were washed three time in PBS and then stained with the In-Situ Cell Death Detection Kit Fluorescein (Sigma) according to the manufacturer's protocol. Cells were washed three times in PBS and Click-iT Plus EdU Alexa Fluor 647 Imaging Kit (ThermoFisher Scientific) was used to detect EdU according to the manufacturer's protocol. Cells were then washed and incubated with Hoechst 33342 for nuclear counterstain.

## Oxidation and reduction of mouse recombinant Igfbp2 (mrIgfbp2)

MrIgfbp2 (R&D Systems) was incubated with 20 mM Dithiothreitol (DTT) for 30 min at 37 °C (Red), 1 mM H$_2$O$_2$ for 10 min at room temperature (Oxi) or with DTT for 30 min followed by H$_2$O$_2$ for 10 min (Red→Oxi). All incubations were performed under Argon and each incubation was followed by removal of excess DTT or H$_2$O$_2$ using Zeba spin desalting columns (ThermoFisher). DTT-, H$_2$O$_2$-, or DTT→H$_2$O$_2$-Pretreated mrIgfbp2 was then added to *Ncf1*[−/−] NSC cultures.

## Assessment of Igfbp2 binding affinity to Igf1 and Igf2

Binding affinity studies were performed as previously described[46] with the following modifications, 1 ng of native, reduced, or oxidized mouse rIgfbp2 was incubated with 10, 30, 100, or 300 ng/ml of Biotinylated rat rIgf2 or mouse rIgf1 (Eagle Bioscience) in a total volume of 500 μl of 50 mM Tris buffer containing 1% BSA pH 7.4 for 22 h at 4 °C. 100 μl of each reaction was added to an ELISA well coated with rat anti-Igfbp2 antibody (R & D Systems). Detection of Igfbp2-bound biotinylated Igfs was done using HRP-conjugated streptavidin followed by TMB. Uncoated wells, wells coated with rat anti-Igfbp2 but without the addition of the binding reaction, and coated wells with the addition of biotinylated Igfs without Igfbp2 were used as controls and for

normalization. The experiment was performed twice with two replicates for each condition/experiment.

## Collection of protein from NSC conditioned medium and LC-MS/MS secretome analysis

P2 NSCs from one WT and one $Ncf1^{-/-}$ pup were seeded at a density of 100,000 cells ml$^{-1}$ in T-25 flasks (Thermo) and maintained under proliferation conditions for 5 days. Conditioned medium was collected, centrifuged, filtered (0.22 μm pores), and flash frozen in liquid nitrogen and stored at −80 °C. The concentration of protein in the conditioned medium was measured using the BCA Protein assay (ThermoFisher Scientific). Samples were lyophilized using a speed-vac, denatured, and solubilized in 50 μl of 8 M urea in 0.1 M triethyl ammonium bicarbonate (TEAB). Samples were then reduced in 10 mM dithiothreitol (DTT) for 1 h at 37 °C and alkylated in 55 mM iodoacetamide (IAA) for 1 h at room temperature in the dark. Alkylated proteins from WT and $Ncf1^{-/-}$ NSCs were then digested in trypsin MS grade at a 1:50 ratio overnight at 37 °C. Digested peptides were then desalted using C18 Microspin columns (Nest group). Digested peptides were run on LTQ Liquid chromatography mass spectrometer (Thermo). The resultant traces were searched against mouse IPI database using Mascot search engine for protein identification to generate mascot generic (mgf) files. Mgf files were then used in Scaffold (Proteome Software Inc., Portland, OR) to visualize and compare the secretomes from WT and $Ncf1^{-/-}$ NSCs.

## Collection of protein from conditioned medium, preparation of total cell lysates, and western blot analysis

P2 NSCs from each pup were seeded at a density of 100,000 cells ml$^{-1}$ in T-25 flasks (Thermo) and maintained under proliferation conditions for 5 days. Neurospheres suspended in conditioned medium were harvested in a 15 ml conical tube and centrifuged. Cell pellets were lysed in RIPA buffer (Sigma) supplemented with Pierce phosphatase inhibitor mini tablets (Pierce) and cOmplete protease inhibitor cocktail (Roche) on ice for 20 min and sonicated for 20 s. Lysates were centrifuged at 13,000 × g for 5 min at 4 °C. Supernatant was collected in screw capped 2 ml tubes, snap frozen in liquid nitrogen, and stored at −80 °C. Conditioned medium was further centrifuged and filtered (0.22 μm pores), snap frozen in liquid nitrogen, and stored at −80 °C. The concentration of protein in the conditioned medium was measured using the Bio-Rad Protein assay (Bio-Rad). For each sample, 150 μg of protein was loaded onto a 15% or 4–20% gradient SDS polyacrylamide gel after boiling in Laemmli buffer (Sigma). Protein bands were transferred to a nitrocellulose membrane, which was then blocked in 2% bovine serum albumin (BSA) in filtered 1 x TBS buffer (36 mM Tris Base, 50 mM NaCl and 0.5% Tween 20). The membrane was then incubated with rabbit polyclonal anti-Igfbp2 (Millipore, 1:1000), goat polyclonal anti-Igfbp2 (R & D systems, 1:500), rabbit monoclonal anti-γ-H2ax (Abcam, 1:1000), goat polyclonal anti-Gapdh (ThermoFisher, 1:1000), Rad51 (mouse, 1:1000, Abcam), Nestin (chicken, 1:750, Novus Biologicals), or mouse monoclonal anti-Flag (Sigma, 1:500) in 0.5% BSA in filtered 1 x TBS, overnight at 4 °C. The membrane was then washed and incubated in 0.5% BSA in filtered 1 x TBS containing IRDye 680 donkey-anti-rabbit, IRDye 800 donkey-anti-goat antibody (1:10,000, LI-COR Biosciences), Alexa Fluor 488-Conjugated Donkey anti-chicken (Jackson ImmunoResearch, 1:10,000), or HRP-Conjugated Donkey anti-mouse (Jackson ImmunoResearch, 1:10,000) for 2 h at room temperature. The membrane was then imaged using the Odyssey or Odyssey M imaging system (LI-CORE Biosciences). Densitometry was carried out using the Image Studio Lite software and Empiria Studio (LI-CORE Biosciences).

## RNA sequencing and pathway analysis

Primary NSCs were prepared from 4 WT and 4 $Ncf1^{-/-}$ pups. P2 NSCs from each pup were seeded at a density of 100,000 cells ml$^{-1}$ in T-25 flasks (Thermo) and maintained under proliferation conditions. At 48 h after plating, paired $Ncf1^{-/-}$ NSC cultures from each animal were treated with vehicle or Igfbp2 (60 ng ml$^{-1}$). This concentration of Igfbp2 was calculated based on the abundance of Igfbp2 in the media of WT culture. Forty-eight hours later, total RNA was prepared. Samples were depleted of rRNA and RNAseq was performed on the remaining RNA by the Genomics Core at the University of Washington. Median quality for all bases was equivalent for all samples; the median Phred quality scores for each base was ≥32. Samples were mapped using the Bowtie2 software and quantified with RSEM[47]. All downstream analyses used transcripts per million (TPM) as units. One-way ANOVAs was performed using the aov function in R version 3.2.1. Tukey's post hoc test was then performed for all genes using the TukeyHSD function. All $p$-values were then FDR corrected by the Benjamini−Hoeschberg (BH) correction using the p.adjust function in R. All subsets of genes were selected using the BH FDR corrected q-values, and cutoffs of 0.05. All heat maps were generated using the heatmap.2 function, using Euclidean distance from the gplots library in R. For heat map display, all genes were log transformed [log2(TPMi+1)] and then linear normalized [logTPMi − min(logTPM)]/[max(logTPM) − min(logTPM)]. The minimal and maximal values used were the minimum and maximum TPM for each gene. Biological pathway analysis was performed using Integrated Pathway Analysis software (Ingenuity Systems, Redwood City, CA) and the absolute-fold change in gene expression for the indicated comparisons. The positive pathway identification threshold was set to $P < 0.05$ by Fisher's exact test. Ingenuity Pathway Analysis (IPA) software (Qiagen) was used to determine GO terms and calculate the differentially activated or inhibited pathways and biological functions in the $Ncf1^{-/-}$ transcriptomes compared to wild-type counterpart.

## siRNA knockdown experiments

200,000 NSCs were plated into all wells of 6-well dishes and allowed to grow for 3 days prior to transfection with dicer substrate interfering RNA (dsiRNA). dsiRNA oligos were ordered from IDT, resuspended in 200 μl/nanomole Invitrogen ultrapure water, and used to transfect cells (with Lipofectamine RNAiMAX, Invitrogen). For each well in a 6-well dish, 5 μl of dsiRNA stock (25 nM final concentration) was mixed with 125 μl of OptiMEM, while separately 10 μl of Lipofectamine RNAiMAX was mixed with 125 μl OptiMEM; these fractions were then mixed and incubated at room temperature for 5 min, then added drop-wise to wells. The IDT NC1 dsiRNA was used as a negative control. *Fanca, Fancd2,* or *Rad51* were knocked down using the following IDT dsiRNA kits: mm.Ri.*Fanca*.13.1, mm.Ri.*Fancd2*.13.1, or mm.Ri.*Rad51*.13.1.

## Mouse *Fanca* (*mfanca*)-PiggyBac vector plasmid construction

NCBI gene number 14087 was used to synthesize *mFanca* gene flanked by XbaI at the 5′-end and NotI at the 3′-end inserted in pUC57 (Genscript). The *mFanca* gene fragment was subcloned into the XbaI and NotI restriction enzyme sites downstream of the CMV promoter in the PiggyBac vector plasmid, which also harbored a puromycin-resistant gene cassette (cat. # PB513B-1; SBI System Biosciences). A negative control PiggyBac vector containing GFP and puromycin-resistant gene expression cassettes was also generated.

## Generating *mFanca*-overexpressing (Fanca OE) mouse NSCs

Passage 2 WT NSCs were transfected with Super piggyBac Transposase expression vector (Cat. # PB210PA-1; SBI System Biosciences) in addition to *mFanca* or GFP-expressing PiggyBac vector plasmid using the Lipofectamine LTX with Plus Reagent (Invitrogen). Transduced NSCs were selected in 3 μg ml$^{-1}$ puromycin for 1 day.

## Analysis of Igfbp2 cysteine oxidation

Two protocols were used for oxidation and reduction of redox-sensitive cysteine residues of Igfbp2. For oxidation, mrIgfbp2 ($120\,\mu g\,ml^{-1}$) was incubated with 0 μM, 10 μM, 100 μM, or 1 mM $H_2O_2$ at pH 7 for 10 min at room temperature. This treatment typically oxidizes the redox-sensitive thiol residues to form disulfide bonds. In this protocol, excess $H_2O_2$ was eliminated by incubation with catalase ($0.1\,\mu g\,ml^{-1}$) for 15 min at room temperature. For reduction, mrIgfbp2 (20 μM) was incubated with a 4 M excess of tris[2-carboxyethyl] phosphine (TCEP) (80 μM) in Sodium Citrate for 15 min at room temperature. This treatment permanently reduces all the oxidized redox-sensitive cysteine residues to thiol form. 1% Acetic acid was added to adjust the pH to 6.5. The results of both protocols were then incubated in N-(Biotinoyl)-N′-(iodoacetyl)ethylenediamine (BIAM) (100 μM) for 15 min at room temperature. This step typically labels all the available thiol groups with BIAM, which is detectable by WB and LC-MS/MS. Excess BIAM was quenched by adding β-Mercaptoethanol (BME) to a final concentration of 20 mM. The pH was adjusted to 8.5 using 1.2 M Tris base buffer. Subsequently, 6 μl of 6x Laemmli buffer were added to 30 μl of each reaction and the samples were boiled. Equal amounts of each reaction were subjected to SDS-PAGE and protein bands were transferred to a nitrocellulose membrane. Immunoblotting was then performed using IR 800-conjugated streptavidin. The rest of each reaction was submitted to the proteomics core at the University of Iowa Carver College of Medicine for LC-MS/MS analysis and evaluation of the BIAM-labeled cysteine residues of Igfbp2.

## Generating mutant Igfbp2 plasmids

We used the pCMV-mIgfbp2 plasmid, which carries a Myc-DDK-tagged mouse Igfbp2 cDNA (NM_008342) (Origene). To generate the pCMV-Igfbp2$^{C263A}$ plasmid, a partial cDNA fragment of mouse Igfbp2 harboring the C263A mutation was synthesized (IDT, Coralville, IA) and exchanged into pCMV-mIgfbp2. To generate the pCMV-Igfbp2$^{C43A}$ and pCMV-Igfbp2$^{C43/263A}$ plasmids, we used the forward primer CGCTGC CCACCC*GCC*ACGCCCGAGCG and reverse primer CGCTCGGGCGT *GGC*GGGTGGGCAGCG with the QuikChange II XL Site-Directed Mutagenesis Kit (Agilent Technologies) according to manufacturer's instructions.

## Expression of WT and mutant mIgfbp2 in *Igfbp2*$^{-/-}$ NSCs

*Igfbp2*$^{-/-}$ NSCs were harvested from newborn pups and 500,000 cells were plated onto each well of 6-well dishes. Cells were allowed to grow for 24 h prior to transfection with plasmids carrying the WT or mutant mouse Igfbp2 cDNA. Cells were transfected using the Lipofectamine LTX with Plus Reagent (Invitrogen). For each well in a 6-well dish, 5 μg of plasmid DNA stock was mixed with 150 μl of OptiMEM and 5 μl of Plus Reagent, while separately 12 μl of Lipofectamine LTX was mixed with 150 μl OptiMEM. These fractions were then mixed and incubated at room temperature for 5 min, then added drop-wise to wells. We used pCMV-Igfbp2 (Origene), pCMV-Igfbp2$^{C43A}$, pCMV-Igfbp2$^{C263A}$, or pCMV-Igfbp2$^{C43/263A}$ plasmids for transfections. pCMV6-eGFP plasmid was used as a negative control. Eighteen hours after the start of transfection, the medium was replaced with a mixture of fresh growth medium and conditioned medium at a ratio of 2:1.

## Reverse transcription and quantitative polymerase chain reactions (rtPCR and qPCR)

P3 *Igfbp2*$^{-/-}$ NSCs were plated onto all wells of 6-well dishes. Three days after transfection with the indicated plasmids, 200 μl of conditioned medium were collected for western analysis, to ensure that WT and mutant Igfbp2 were expressed and secreted. After a further 48 h, the cells were harvested, and total RNA was prepared using the RNeasy Plus Mini Kit (Qiagen). cDNA was prepared from the total RNA using the High Capacity cDNA Reverse Transcription Kit (Applied Biosystems) or SuperScript IV VILO Master Mix (ThermoFisher Scientific) according to the manufacturer's protocol. Quantification of transcription levels relative to β-Actin (inner control) was performed using TaqMan Universal Master Mix II, with UNG (Applied Biosystems), according to the manufacturer's protocol and utilizing the following gene expression assays: Mm02619580_g1 Actb 4448484 TaqMan Gene Expression Assay, SM VIC PL for β-Actin, Mm01243365_m1 Fanca 4331182 TaqMan Gene Expression Assay, SM for Fanca, Mm01184611_m1 Fancd2 4331182 TaqMan Gene Expression Assay, SM for Fancd2 and Mm00487905_m1 Rad51 4331182 TaqMan Gene Expression Assay, SM for Rad51.

## Imaging and image processing

All immunofluorescence (IF) images were acquired using a laser scanning confocal microscope LSM-700 (Zeiss; ×20 objective), LSM-880 (Zeiss, ×20 and ×63 objectives), and LSM-980 (Zeiss, ×20 and ×63 objectives) in conjunction with the acquisition software Zen-2010. Tiled scans of the areas of interest in the mouse brain and of all neurospheres in the glass-bottom well of a 35-mm plate were acquired. Image processing and quantification were performed using offline MetaMorph software (Molecular Devices). Quantification of the numbers of cells positive and negative for specific markers or a thymidine analog was carried out using the multi-wavelength cell scoring application module of the MetaMorph software. This module allowed for the following parameters to be set: the average size of the nucleus and cytosol; whether a marker is expressed in the nucleus, cytosol, or both; and the intensity above local background and the percentage of the cell that must be stained for it to be considered positive for a certain marker. All bright field (BF) images were captured using spinning disc DMI 6000 (Leica, ×2.5, ×10, and ×20 objectives) in conjunction with the acquisition software online MetaMorph. Tiled scans of the whole wells or flasks were acquired. Image processing and quantification of neurosphere number and dimensions were performed using offline MetaMorph software.

## Statistical analysis

Unless otherwise specified, statistical analysis and the calculation of significance values were performed using the Prism 7 (GraphPad) software. The two-tailed Mann–Whitney U test, two-tailed student's t-test, as well as Kruskal–Wallis, one-way and two-way ANOVA, were used when appropriate. The Bonferroni or Benjamini–Hochberg FDR post-test comparing all groups was used where indicated; significance was considered at $p < 0.05$. Exact $p$ values and details of each statistical test are available in the accompanying Source data file. Error bars indicate s.e.m. All bar graphs were assembled using Excel (Office 365).

## Reporting summary

Further information on research design is available in the Nature Portfolio Reporting Summary linked to this article.

# Data availability

The RNA sequencing data generated in this study has been deposited in the NCBI Gene Expression Omnibus (GEO) (https://www.ncbi.nlm.nih.gov/geo/) under accession code GSE221955. The main data supporting the findings of this study are available within the article and its Supplementary Figures and supplementary data files. Complete statistics and exact $P$ values for each test and post hoc comparison are also included within the Source data file. Source data are provided with this paper.

# Code availability

The customized code used for RNAseq data analysis is attached as Supplementary Software 1.

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

## Acknowledgements

The authors thank Clifford J. Rosen for his kind gift of the *Igfbp2*⁻/⁻ mice, Nicholas J. Pantazis, and Douglas Spitz for their valuable discussion and input into the project. We also gratefully acknowledge: The Carver Trust for funding for this research through the Endowed Chair for Molecular Medicine to J.F.E.

## Author contributions

W.S.S, S.O.E., B.M., and Y.Z. performed all in vivo experiments. W.S.S. performed all microscopy imaging, NSC studies, data quantification, and pathway analysis. S.R.T. performed the bioinformatic RNAseq analysis and statistics and contributed to data analysis and manuscript editing. S.O.E. performed genotyping and mouse colony care. W.S.S., S.O.E., S.H.C., and W.Z. performed all in vitro experiments, western blotting, and ELISA. W.S.S. and T.I.E. ran LC–MS/MS experiments. C.Y. contributed to the study conceptualization and manuscript editing. W.S.S. and J.F.E. designed all studies, analyzed the data, and wrote the manuscript.

## Competing interests

The authors declare no competing interests.
