## [Peer Review File · Nature Communications]

Redox-Dependent Igfbp2 Signaling Controls Brca1 DNA Damage Response to Govern Neural Stem Cell FateReviewers' comments:

Reviewer #1 (Remarks to the Author):

Overall there are some interesting findings in this study associated with the impact of loss of Ncf1 in NSCs. However, some of the presented data is phenomenological with conclusions that are unclear/equivocal. The claim that inactivating DNA repair promotes neurosphere forming ability (i.e. similar to WT cells) is surprising and it's difficult to understand why inhibiting key genome stability factors and increasing DNA damage (shown by γ H2AX formation) would promote recovery and be beneficial in terms of NSC function. While this might be a possibly interesting finding, as it stands there is too much uncertainty about what it means and how it happens. The data presented doesn't address how regulation of repair pathways modulates NSC function, if its physiologically relevant and the mechanistic nature of what DNA damage does to alter cell fate. The regulation of stem cell self-renewal or lineage commitment via redox regulation may well underpin much of what is going on, but Redox regulation potentially targets multiple cellular proteins/pathways and the connection to DNA damage may not be central to this process. For instance, although data is shown for some FA pathway members, other genes identified such as ATRIP and others will substantially compromise cellular growth/viability. Moreover, compromising repair and increasing DNA breaks as a means to regulate NSC maintenance and lineage commitment seems at odds with the need to maintain genome stability to prevent mutations in these progenitors.

Reviewer #2 (Remarks to the Author):

This is a very interesting study in which Shahin and colleagues show that redox regulation of Igfbp2 controls Brca1 DNA damage to govern neural stem cell fate. To achieve such a conclusion, the authors used a genetic mouse model lacking the NADPH oxidase component Ncf1, hence significantly reducing the level of endogenous ROS formation. By controlling endogenous ROS production, the authors found the redox-mediated modulation of Igfbp2, identified the Cys residues responsible for this effect, and the consequent regulation of DNA repair in NSCs impacting on NSC fate decisions.

The work is nicely designed and elegantly performed. However, this reviewer has found a couple of minor issues that the authors should consider taking them into account in order to, maybe, improving the impact of the message.

1. Knocking out a NADPH oxidase subunit might have other consequences besides the reduction in superoxide formation. For instance, it is known that, by producing superoxide, NADPH oxidase uncouples eNOS activity in endothelial cells (doi: 10.1007/s00125-012-2557-6), a phenomenon that enhances superoxide production, at the expense of reduced NO production, by eNOS (DOI: 10.1073/pnas.95.16.9220). Therefore, Ncf1 KO cells might

have a reduction in ROS because of the recoupling of eNOS (or, in the case of NSC, nNOS) that results in more NO formation at the expense of superoxide formation. If so, one might speculate that the effect of Ncf1 KO in NSCs fate might be due to, not simply to reduced ROS, but to increased NO, which is dependent on nNOS, subjected to the same kind of uncoupling as eNOS (PMID: 1280257). In fact, in NSC, nNOS-derived NO is required for NSC differentiation (doi: 10.3389/fncel.2017.00066; PMID: 7513691). Therefore, to make a stronger assertion that it is ROS, and not NO, the molecule(s) responsible for the observed effects (at least some of them) upon endogenous modulation of Ncf1, it might be interesting to modulate the NO-forming nNOS activity using, e.g., nNOS specific inhibitors.

2. The other concern is related to the use of the ROS probe hydroethidine. This is a rather unspecific probe that, actually, is unable to distinguish amongst different types of reactive oxygen and nitrogen species. This is not necessarily a problem for the conclusions of the work. However, the authors might consider double checking that it is superoxide/H₂O₂ the molecules involved in the regulation of the Cys residues, and not NO (which can also nitrosylate -SH groups). One possibility would be to monitor NO formation or detecting a NO-derived footprint.

Juan P Bolanos

Reviewer #3 (Remarks to the Author):

The NADPH oxidase complex (NOX/NCF1) is a major source for the generation of reactive oxygen species. The manuscript "Redox-Dependent Igfbp2 Signaling Controls Brca1 DNA Damage Response to Govern Neural Stem Cell Fate" by Shahin and colleagues investigates the function of the NADPH oxidase complex in the control of neural stem cells in the adult brain. To this end the authors examine Ncf1 knockout mice. Here they show that Ncf1 knockout is associated with decreased Superoxide levels in the neurogenic zones of the adult murine brain. BrdU pulse chasing indicates increased proliferation in one of the neurogenic zones, that is the subventricular zone / rostral migratory stream, of Ncf1 knockout mice. The authors then investigate the Ncf1-deficient neural stem cells derived from day 1 / day 2 postnatal brains. Ncf1-deficient neural stem cells rather than forming neurospheres generated sheet-like structures and generated increased numbers of oligodendrocytes and decreased numbers of neurons. The Ncf1 knockout phenotype could be rescued by a secreted factor generated by WT neural stem cells. The authors provide evidence that this factor is IGFBP2 and that the redox-state of the cysteine residue C43 of IGFBP2 is critical for the ability of IGFBP2 to rescue the Ncf1 ko neural stem cell phenotype. Using transcriptomic analysis the authors find that Ncf1 ko is associated with higher activity

of the BRCA1-dependent DNA repair system. shRNA mediated knockdown of different components of this system normalized Ncf1ko neural stem cell behavior. Based on these findings, the authors conclude that the NOX/NCF1 – ROS – IGFBP2 – BRCA1 axis regulates neural stem cell proliferation and differentiation.

ROS have previously been implicated in the regulation of neural stem cell function in the adult mammalian brain. The novelty of this study would be the delineation of a new pathway NOX/NCF1 – ROS – IGFBP2 – BRCA1 in this process. Such identification would per se be quite interesting for the field, in particular because of the surprising observation that overactivity of the BRCA1 pathway is associated with dysregulation of neural stem cell behavior and decreased generation of neurons. I have, however, a number of major concerns with the present manuscript:

1. The authors describe that Ncf1ko increases proliferation, increases oligodendrogenesis and impairs neurogenesis in vitro. To strengthen the relevance of the in vitro data the authors should investigate determine the generation of oligodendrocytes and neurons in the SVZ in vivo. In general strengthening the evidence for the in vivo relevance of the NOX/NCF1 – ROS – IGFBP2 – BRCA1 axis in neural stem cell regulation would be important.
2. In vivo data were collected in adult mice whereas in vitro data stem from postnatal day 1 / 2 neural stem cell cultures. As it is unclear whether early postnatal stem cells differ substantially from adult neural stem cells the authors should perform their assays also in adult mouse derived cultures.
3. The cell biological mechanism leading to the sheet like structures and shifts in the generation of oligodendrocytes and neurons is only superficially examined. The authors suggest that the phenotype represents a proliferation and fate determination phenotype but there is no direct assessment of proliferation e.g. via BrdU incorporation, fraction of proliferation marker expressing cells. Cell death as a contributor to the shift in oligodendrocyte and neuron generation has to be examined.
4. The authors suggest that the redox state of IGFBP2 determines the release of IGFBP2. In the experiments described in lines 79-85 the authors find that supplementing native and oxidized IGFBP2 but not reduced IGFBP2 induced neurosphere formation of Ncf1 knockout cells. As the proteins are supplemented to the medium, impaired release of reduced IGFBP2 cannot be the reason that reduced IGFBP2 does not rescue the phenotype. How do the authors explain this observation? Does reduced IGFBP2 have different affinities for IGFs?
5. A major argument for the Ncf1/Igfbp2 link is the observation that Igfbp2 ko stem cells behave similar to Ncf1 ko stem cells. Ncf1 ko stem cells can be rescued by treatment with H2O2. How does H2O2 affect Igfbp2 ko cells? This is a critical control experiment to establish the importance of the NOX/NCF1 – ROS – IGFBP2 axis.
6. The authors report that knockdown of BRCA1 pathway components rescues the Ncf1 proliferation and differentiation phenotype. Please provide evidence for the knockdown efficiency. As pointed out under comment 3) the authors should perform more direct assays to clearly distinguish the contributions of proliferation, fate determination and cell death.
7. Line 129 -130: LC-MS/MS analysis of these BIAM-treated Igfbp2 substrates identified two cysteines, C43 and C263, as the most redox-sensitive residues (data not shown). This is an interesting and important data point, please show the data.

8. Please examine DNA damage / evidence for double strand breaks also in the in vivo context.

9. The very surprising observation is the potential positive effect of DNA damage to stimulate the generation of neurons. It would be interesting to support this most interesting finding by examining the impact of knockout of BRCA1 pathway components on neurogenesis in wildtype stem cells.

Reviewers' comments:

Reviewer #1 (Remarks to the Author):

General: Overall there are some interesting findings in this study associated with the impact of loss of Ncf1 in NSCs. However, some of the presented data is phenomenological with conclusions that are unclear/equivocal.

General Response: We thank the reviewer for bringing forward the lack of clarity of some conclusions. We have rewritten the manuscript to present data in a hypothesis-driven way and have been able to make clearer conclusions with data from new suggested experiments.

Q1. The claim that inactivating DNA repair promotes neurosphere forming ability (i.e. similar to WT cells) is surprising and it's difficult to understand why inhibiting key genome stability factors and increasing DNA damage (shown by γ H2AX formation) would promote recovery and be beneficial in terms of NSC function.

R1. We do not propose that excessive DNA damage is “beneficial” to NSCs, but rather that maintaining WT levels of DNA damage may be required for checkpoint exit from a self-renewing state during the commitment to form neurospheres. Our new *in vivo* data on γ -H2ax and Rad51 expression in the SVZ of *Ncf1*^{-/-} and *Igfbp2*^{-/-} mice also support the findings in NSCs (Supplementary Fig. 9a–b). Here we will review data that supports the need for DNA damage as an integral component of NSC lineage commitment. First, *Ncf1*^{-/-} NSCs have higher levels of Brca1 pathway genes than WT NSCs. We believe this results in lower levels of DDSBs (less γ -H2ax) and a protective cellular state that drives self-renewal of NSCs (Fig. 1c–e). Second, transiently reducing *Fanca*, *Fancd2* and *Rad51* transcript levels in *Ncf1*^{-/-} NSCs with dsiRNA (Supplementary Fig. 7a–c) leads to the induction of low levels of γ -H2ax (Fig. 3i–j) and neurosphere formation (Fig. 3e–f and Supplementary Fig. 8a & c) and returns EdU incorporation and cell death back to WT levels (Supplementary Fig. 9d–f). It is important to point out that this is a transient downregulation of the Brca1 pathway genes and likely in only a subpopulation of NSCs. Permanent inhibition of Brca1 pathway genes using Cas9 or lentiviral shRNA approaches would certainly be expected to be toxic to NSCs. We are suggesting that NSCs must transiently downregulate DNA repair pathways to exit the self-renewing state and the consequence of this process leads to enhanced DDSBs. Whether DNA damage is a required event for NSC commitment toward more differentiated fates, or simply a consequence of the commitment process, has not been formally addressed.

Previous published studies suggest that both neural stem cells (NSCs) (Wei, Chang et al. 2016) and neurons (Madabhushi, Gao et al. 2015) maintain a certain level of DNA damage required

for proper function. We show that reducing the extent of DNA damage as a result of low levels of ROS or loss of Igfbp2, in *Ncf1*^{-/-} and *Igfbp2*^{-/-} respectively, resulted in faster proliferation of NSCs and less commitment toward downstream neural progenitors and thus impairs neurosphere formation (Fig. 1a–c) and cell fate decisions (Supplementary Fig. 4a–e, Supplementary Fig. 5e–h and Supplementary Table 2c). This phenotype was rescued by knockdown of *Fanca*, *Fancd2* or *Rad51* (Fig. 3e–f, and Supplementary Fig. 4j–k).

Q2. While this might be a possibly interesting finding, as it stands there is too much uncertainty about what it means and how it happens. The data presented doesn't address how regulation of repair pathways modulates NSC function, if its physiologically relevant and the mechanistic nature of what DNA damage does to alter cell fate. The regulation of stem cell self-renewal or lineage commitment via redox regulation may well underpin much of what is going on, but Redox regulation potentially targets multiple cellular proteins/pathways and the connection to DNA damage may not be central to this process.

R2. We appreciate the fact that the functional requirement for DNA damage in NSC fate decisions has not been determined, indeed we discuss this in the text. We also appreciate that ROS has the potential to act on multiple pathways that could impact NSC behavior. However, we now show new data that H₂O₂ treatment of *Igfbp2*^{-/-} NSCs did not restore neurosphere formation (Fig. 2g–h) nor reduce transcript levels of *Fanca*, *Fancd2* or *Rad51* to WT levels (Supplementary Fig. 7g–i), indicating that Igfbp2 is the downstream effector that mediates the regulation of NSCs through *Fanca*, *Fancd2* and *Rad51*. Furthermore, we have shown that adding back native or oxidized, but not reduced Igfbp2, to *Ncf1*^{-/-} NSCs restored neurosphere formation (Fig. 2c–d). Additionally, transient knockdown of *Fanca*, *Fancd2* or *Rad51* restored neurosphere formation (Fig. 3e–f), elevated γ -H2ax to WT levels (Fig. 3i–j) and rescued differentiation profile of *Ncf1*^{-/-} NSCs (Supplementary Fig. 4j–k). Thus, we believe our study does demonstrate the importance of redox-dependent Igfbp2 signaling and its impact on DNA repair networks that control NSC behavior, and that modulating the DNA repair genes themselves can accomplish the same events. Whether DNA damage is a required component of this mechanism remains to be determined. While it is interesting to speculate DNA breaks may control NSC function based on published studies (cited in R1), it is equally plausible that these changes induce cell cycle checkpoints conducive to NSC fate alterations. We have attempted to better outline these possibilities in the revised text.

Q3. For instance, although data is shown for some FA pathway members, other genes identified such as ATRIP and others will substantially compromise cellular growth/viability.

R3. To address the concern of compromised cell growth/viability, we assessed EdU incorporation and cell death of WT and *Ncf1*^{-/-} NSCs. We now show EdU incorporation and cell death data for WT and *Ncf1*^{-/-} NSCs and this confirms higher proliferation and slightly decreased cell death in *Ncf1*^{-/-} NSCs compared to WT ones (Fig. 1d–e). See R2 for more details.

Q4. Moreover, compromising repair and increasing DNA breaks as a means to regulate NSC maintenance and lineage commitment seems at odds with the need to maintain genome stability to prevent mutations in these progenitors.

R4. Indeed, maintaining genome stability and avoiding accumulation of mutations is important for maintenance of NSCs. However, both NSCs (Wei, Chang et al. 2016) and neurons (Madabhushi, Gao et al. 2015) have been previously shown to maintain a certain level of DNA double strand breaks (DSBs) which might be required for proper function. Our study shows

that normal levels of DNA repair genes and DDSBs are required for NSCs to exit the proliferative and self-renewing state and give rise to various neural progenitors. See R2 for more details.

Reviewer #2:

General: This is a very interesting study in which Shahin and colleagues show that redox regulation of Igfbp2 controls Brca1 DNA damage to govern neural stem cell fate. To achieve such a conclusion, the authors used a genetic mouse model lacking the NADPH oxidase component Ncf1, hence significantly reducing the level of endogenous ROS formation. By controlling endogenous ROS production, the authors found the redox-mediated modulation of Igfbp2, identified the Cys residues responsible for this effect, and the consequent regulation of DNA repair in NSCs impacting on NSC fate decisions. The work is nicely designed and elegantly performed. However, this reviewer has found a couple of minor issues that the authors should consider taking them into account in order to, maybe, improving the impact of the message.

General response: We would like to thank the reviewer for his constructive comments.

Q1a. Knocking out a NADPH oxidase subunit might have other consequences besides the reduction in superoxide formation. For instance, it is known that, by producing superoxide, NADPH oxidase uncouples eNOS activity in endothelial cells (doi: 10.1007/s00125-012-2557-6), a phenomenon that enhances superoxide production, at the expense of reduced NO production, by eNOS (DOI: 10.1073/pnas.95.16.9220). Therefore, Ncf1 KO cells might have a reduction in ROS because of the recoupling of eNOS (or, in the case of NSC, nNOS) that results in more NO formation at the expense of superoxide formation. If so, one might speculate that the effect of Ncf1 KO in NSCs fate might be due to, not simply to reduced ROS, but to increased NO, which is dependent on nNOS, subjected to the same kind of uncoupling as eNOS (PMID: 1280257). In fact, in NSC, nNOS-derived NO is required for NSC differentiation (doi: 10.3389/fncel.2017.00066; PMID: 7513691).

R1a. We appreciate the helpful criticism and suggestion of the reviewer. Indeed, the possible involvement of nNos-generated NO in the process is intriguing based on previous studies cited by the reviewer. The reviewer hypothesized that absence of Ncf1 results in recoupling of nNos, producing more NO which might be responsible for downstream effects through increased Nitrosylation. If this holds true, we would expect to see an increase in nitrosylation in the SVZ of *Ncf1*^{-/-} compared to WT and *Igfbp2*^{-/-}. We carried out biotin derivatization to detect cysteine S-Nitrosylation in the SVZ of WT, *Ncf1*^{-/-} and *Igfbp2*^{-/-} mice. Interestingly, we saw the opposite of what we expected. The SVZ of *Ncf1*^{-/-} mice showed *lower* cysteine S-Nitrosylation compared to WT and *Igfbp2*^{-/-} counterparts. This might be explained by the fact that NO reacts with superoxide anions to form the potent oxidant peroxynitrite (ONOO⁻) that nitrosylates SH groups (Forstermann and Sessa 2012) or perhaps, nNos is more active in presence of Ncf1.

As the reviewer points out, nNos-generated NO is required for NSC neural fate commitment (Jin, Yu et al. 2017). The above *in vivo* results of reduced cysteine S-Nitrosylation in the SVZ of *Ncf1*^{-/-} mice and our finding that *Ncf1*^{-/-} NSCs show less neurogenesis than WT counterparts (Supplementary Fig. 4a–h), is consistent with published NO involvement in neurogenesis. However, an NO-centric mechanism for our findings cannot explain why there are normal levels of cysteine S-Nitrosylation in the SVZ of *Igfbp2*^{-/-} mice, yet both *Ncf1*^{-/-} and *Igfbp2*^{-/-} NSCs demonstrate reduced neurogenesis as compared to WT NSCs. Furthermore, L-VNIO-treated

Ncf1^{-/-} and *Igfbp2*^{-/-} NSCs showed no changes in *Fanca*, *Fancd2* or *Rad51* transcript levels (Supplementary Fig. 7d–i). Thus, while we recognize the importance of NO on NSC differentiation, we conclude NO acts through pathways that does not involve *Igfbp2*.

Q1b. Therefore, to make a stronger assertion that it is ROS, and not NO, the molecule(s) responsible for the observed effects (at least some of them) upon endogenous modulation of Ncf1, it might be interesting to modulate the NO-forming nNOS activity using, e.g., nNOS specific inhibitors.

R1b. We again thank the reviewer for suggesting this experiment and we have performed new experiments to address this interesting hypothesis. As suggested, we used the nNos inhibitor L-VNIO (100 μM) to modulate this pathway in NSCs. L-VNIO-treated *Ncf1*^{-/-} NSCs showed lower proliferation than vehicle-treated controls, with no change in cell death (Fig. 1d–f). Furthermore, L-VNIO-treated *Ncf1*^{-/-} and *Igfbp2*^{-/-} NSCs showed no changes in *Fanca*, *Fancd2* or *Rad51* transcript levels (Supplementary Fig. 7d–i). This suggests that nNos activity might regulate NSCs through a different pathway. We attempted to assess the effect of nNos inhibition, using L-VNIO, on neurosphere formation, but the NSCs did not tolerate prolonged exposure (11 days) to L-VNIO regardless of genotype.

Q2. The other concern is related to the use of the ROS probe hydroethidine. This is a rather unspecific probe that, actually, is unable to distinguish amongst different types of reactive oxygen and nitrogen species. This is not necessarily a problem for the conclusions of the work. However, the authors might consider double checking that it is superoxide/H2O2 the molecules involved in the regulation of the Cys residues, and not NO (which can also nitrosylate -SH groups). One possibility would be to monitor NO formation or detecting a NO-derived footprint.

R2. We would like to thank the reviewer for this comment and we now discuss the results with hydroethidine more conservatively in the manuscript. Our *in vitro* experiments manipulating the redox state of *Igfbp2* most strongly support the fact that cysteine reduction/oxidation (not S-nitrosylation) controls the biologic activity on NSCs. In these experiments we used recombinant mouse *Igfbp2* to treat *Ncf1*^{-/-} NSCs. We pretreated the recombinant *Igfbp2* with Dithiothreitol (DTT) for reduction or H₂O₂ for oxidation before adding it to proliferating NSCs and demonstrated the oxidized form of *Igfbp2* was biologically active (Fig. 2c–d). In a different experiment, we treated recombinant mouse *Igfbp2* protein with tris[2-carboxyethyl] phosphine (TCEP) or H₂O₂ for permanent reduction or oxidation of cysteines, respectively. We then separated proteins via SDS-PAGE, excised relevant bands and ran LC-MS/MS. These studies were useful in identifying the cysteines responsible for biologic activity, which were then studied on NSCs as *Igfbp2* mutants. We have gone back to our LC-MS/MS data and found no evidence for S-Nitrosocysteine peptides in native *Igfbp2* (unreduced or oxidized) produced in a mouse myeloma cell line. We did not use NO donors in our *in vitro* oxidation of *Igfbp2*, so clearly one would not expect S-Nitrosocysteine in those experiments, but the fact that oxidized form of *Igfbp2* (absent LC-MS/MS-confirmed S-Nitrosocysteines) was biologically to induce neurospheres and neurogenesis from *Ncf1*^{-/-} NSCs supports an NO-independent mechanism involving *Igfbp2*. Additionally, the above cited *in vivo* studies (R1a) localizing cysteine S-Nitrosylation in SVZ of WT, *Ncf1*^{-/-} and *Igfbp2*^{-/-} mice do not support a mechanism of NO involvement since *Ncf1*^{-/-} and *Igfbp2*^{-/-} mice have discordant cysteine S-Nitrosylation in the SVZ yet NSCs from these two genotypes have similar *in vitro* phenotypes and similarly altered *in vivo* differentiation profiles in terms of oligodendrogenesis and neurogenesis in the granule cell layer (GCL) of olfactory bulb (Supplementary Fig. 4f–i).

Reviewer #3:

The NADPH oxidase complex (NOX/NCF1) is a major source for the generation of reactive oxygen species. The manuscript "Redox-Dependent Igfbp2 Signaling Controls Brca1 DNA Damage Response to Govern Neural Stem Cell Fate" by Shahin and colleagues investigates the function of the NADPH oxidase complex in the control of neural stem cells in the adult brain. To this end the authors examine Ncf1 knockout mice. Here they show that Ncf1 knockout is associated with decreased Superoxide levels in the neurogenic zones of the adult murine brain. BrdU pulse chasing indicates increased proliferation in one of the neurogenic zones, that is the subventricular zone / rostral migratory stream, of Ncf1 knockout mice. The authors then investigate the Ncf1-deficient neural stem cells derived from day 1 / day 2 postnatal brains. Ncf1-deficient neural stem cells rather than forming neurospheres generated sheet-like structures and generated increased numbers of oligodendrocytes and decreased numbers of neurons. The Ncf1 knockout phenotype could be rescued by a secreted factor generated by WT neural stem cells. The authors provide evidence that this factor is IGFBP2 and that the redox-state of the cysteine residue C43 of IGFBP2 is critical for the ability of IGFBP2 to rescue the Ncf1 ko neural stem cell phenotype. Using transcriptomic analysis the authors find that Ncf1 ko is associated with higher activity of the BRCA1-dependent DNA repair system. shRNA mediated knockdown of different components of this system normalized Ncf1ko neural stem cell behavior. Based on these findings, the authors conclude that the NOX/NCF1 – ROS – IGFBP2 – BRCA1 axis regulates neural stem cell proliferation and differentiation.

ROS have previously been implicated in the regulation of neural stem cell function in the adult mammalian brain. The novelty of this study would be the delineation of a new pathway NOX/NCF1 – ROS – IGFBP2 – BRCA1 in this process. Such identification would per se be quite interesting for the field, in particular because of the surprising observation that overactivity of the BRCA1 pathway is associated with dysregulation of neural stem cell behavior and decreased generation of neurons. I have, however, a number of major concerns with the present manuscript.

General response: We would like to thank the reviewer for the constructive suggestions and useful directions.

Q1. The authors describe that Ncf1ko increases proliferation, increases oligodendrogenesis and impairs neurogenesis in vitro. To strengthen the relevance of the in vitro data the authors should investigate determine the generation of oligodendrocytes and neurons in the SVZ in vivo. In general strengthening the evidence for the in vivo relevance of the NOX/NCF1 – ROS – IGFBP2 – BRCA1 axis in neural stem cell regulation would be important.

R1. We appreciate this important *in vivo* correlate and have now evaluated the extent of oligodendrogenesis and neurogenesis in the granule cell layer (GCL) of olfactory bulb, the final destination of differentiated NSCs in the SVZ. We show that GCL of *Ncf1^{-/-}* and *Igfbp2^{-/-}* adult mice have significantly more oligodendrocytes (O4⁺ cells) and fewer neurons (NeuN⁺ Cells) than WT counterparts (Supplementary Fig. 4f-i). Thus, these *in vivo* data correlate with our *in vitro* finding in differentiated neurospheres.

Q2. In vivo data were collected in adult mice whereas in vitro data stem from postnatal day 1 / 2 neural stem cell cultures. As it is unclear whether early postnatal stem cells differ substantially

from adult neural stem cells the authors should perform their assays also in adult mouse derived cultures.

R2. We appreciate the importance of this comparison and have now prepared adult neural stem cells (ANSCs) from SVZs of WT, *Ncf1*^{-/-}, and *Igfbp2*^{-/-} mice. Similar to neonatal NSCs, ANSCs from WT mice formed neurospheres in culture, whereas *Ncf1*^{-/-} and *Igfbp2*^{-/-} counterparts grew in sheets and rarely made neurospheres (Supplementary Fig. 8a–b). Moreover, treatment with dsRNA to knockdown Fanca, Fancd2 or Rad51, allowed *Ncf1*^{-/-} and *Igfbp2*^{-/-} ANSCs to form neurospheres (Supplementary Fig. 8a–c).

3. The cell biological mechanism leading to the sheet like structures and shifts in the generation of oligodendrocytes and neurons is only superficially examined. The authors suggest that the phenotype represents a proliferation and fate determination phenotype but there is no direct assessment of proliferation e.g. via BrdU incorporation, fraction of proliferation marker expressing cells. Cell death as a contributor to the shift in oligodendrocyte and neuron generation has to be examined.

R3. To evaluate proliferation and the contribution of cell death to NSC lineage commitment changes we see with *Ncf1*^{-/-} NSCs, we have performed EdU incorporation and cell death labeling studies in NSC cultures. These studies demonstrate increased proliferation in *Ncf1*^{-/-} NSCs. However, cell death in *Ncf1*^{-/-} NSCs was not significantly different compared to WT NSCs (Fig. 1d–f).

Q4. The authors suggest that the redox state of IGFBP2 determines the release of IGFBP2. In the experiments described in lines 79-85 the authors find that supplementing native and oxidized IGFBP2 but not reduced IGFBP2 induced neurosphere formation of Ncf1 knockout cells. As the proteins are supplemented to the medium, impaired release of reduced IGFBP2 cannot be the reason that reduced IGFBP2 does not rescue the phenotype. How do the authors explain this observation? Does reduced IGFBP2 have different affinities for IGFs?

R4. Redox-dependent maturation and secretion of Igfbp2 may indeed be linked, given that treating *Ncf1*^{-/-} NSCs with H₂O₂ promoted Igfbp2 secretion and neurosphere formation (Fig. 2a–b). Our *in vitro* reconstitution experiments do not directly determine the link between oxidation of Igfbp2 and its secretion, but they do demonstrate that only the oxidized form of Igfbp2 is bioactive on NSCs to induce neurosphere formation and promote neurogenesis during differentiation. In new studies, neither Igf1 nor Igf2 were detectable by ELISA in the secretome of proliferating NSCs, so direct determination of their binding to Igfbp2 by co-IP was not possible. Nevertheless, we performed Igfbp2 binding assays using recombinant Igf1 and Igf2. Our results show that redox status of Igfbp2 does not change its binding affinity for Igf1 or Igf2 (Supplementary Fig. 10c–d). We suggest that the effect of Igfbp2 on NSCs is Igf-independent since we cannot detect its presence in proliferating NSC cultures and there was no redox-dependence in Igf interaction with Igfbp2.

Q5. A major argument for the Ncf1/Igfbp2 link is the observation that Igfbp2 ko stem cells behave similar to Ncf1 ko stem cells. Ncf1 ko stem cells can be rescued by treatment with H2O2. How does H2O2 affect Igfbp2 ko cells? This is a critical control experiment to establish the importance of the NOX/NCF1 – ROS – IGFBP2 axis.

R5. We express our gratitude to the reviewer for suggesting this important experiment. Indeed, H₂O₂ treatment of *Igfbp2*^{-/-} NSCs had no effect on neurosphere formation (Fig. 2g–h) nor expression of Fanca, Fancd2, or Rad51 (Supplementary Fig. 7g–i). These results support our

conclusion that ROS regulate NSCs through Igfbp2.

Q6. *The authors report that knockdown of BRCA1 pathway components rescues the Ncf1 proliferation and differentiation phenotype. Please provide evidence for the knockdown efficiency. As pointed out under comment 3) the authors should perform more direct assays to clearly distinguish the contributions of proliferation, fate determination and cell death.*

R6. We now show transcript levels of Fanca, Fancd2 and Rad51 after transfection of *Ncf1*^{-/-} NSCs with specific dsRNA (Supplementary Fig. 7a–c). Changes in NSC fate determination following Fanca, Fancd2 and Rad51 knockdown were given in the first version of the manuscript (Now Supplementary Fig. 4i–j). Regarding proliferation and cell death, these studies were performed in WT and *Ncf1*^{-/-} NSCs, see R3. Additionally, knocking down Fanca, Fancd2 or Rad51 returned EdU incorporation and cell death levels in *Ncf1*^{-/-} NSCs to the levels observed in wild type NSCs (Supplementary Fig. 9d–f).

Q7. *Line 129 -130: LC-MS/MS analysis of these BIAM-treated Igfbp2 substrates identified two cysteines, C43 and C263, as the most redox-sensitive residues (data not shown). This is an interesting and important data point, please show the data.*

R7. We now show total ion count (TIC) of BIAM labelled cysteine residues in reduced, native and oxidized states in (Supplementary Fig. 10a–b). Since oxidation of Igfbp2 resulted in formation of neurospheres similar to native Igfbp2, we looked for cysteines that were labelled with BIAM only under reducing conditions.

Q8. *Please examine DNA damage / evidence for double strand breaks also in the in vivo context.*

R8. We now show that SVZ of *Ncf1*^{-/-} and *Igfbp2*^{-/-} show higher percentage of cells positive for Rad51 and fewer γ -H2ax-positive cells than WT counterparts (Supplementary Fig. 9a–c). These findings support our *in vitro* results in NSCs.

Q9. *The very surprising observation is the potential positive effect of DNA damage to stimulate the generation of neurons. It would be interesting to support this most interesting finding by examining the impact of knockout of BRCA1 pathway components on neurogenesis in wildtype stem cells.*

R9. Both NSCs and neurons maintain a certain level of DDSBs. Decreased level of DDSBs as in *Ncf1*^{-/-} and *Igfbp2*^{-/-} NSCs results in increased self-renewal and decreased differentiation. Transient and partial knockdown of Brca1 pathway components resulted in restoring the differentiation capabilities of *Ncf1*^{-/-} NSCs. However, knocking out Fanca increased apoptosis and decreased proliferation and self-renewal of NSCs (Sii-Felice, Etienne et al. 2008) (Sii-Felice, Barroca et al. 2008). Based on these reports, knocking out Brca1 pathway components will likely be lethal to NSCs. We propose DNA damage may be a consequence of checkpoint exit from a self-renewing state during the commitment for form neurospheres and that this exist from a self-renewing state requires a lowering of Brca1 DNA damage response genes. It is important to point out the transient nature of downregulating the Brca1 pathway genes. Transient knockdown of Brca1 pathway components did not alter neurosphere formation in WT NSCs but did enhance neurosphere formation in *Ncf1*^{-/-} and *Igfbp2*^{-/-} NSCs (Supplementary Fig. 8a & c). Permanent inhibition of Brca1 pathway genes using Cas9 or lentiviral shRNA approaches could certainly be deadly to NSCs, as previously published. We are suggesting that NSCs must transient downregulate DNA repair pathways to exit the self-renewing state and the

consequence of this process leads to enhanced DSDBs. Whether DNA damage is a required event for NSC commitment toward more differentiated fates, or simply a consequence of the commitment process, has not been formally addressed.

- Forstermann, U. and W. C. Sessa (2012). "Nitric oxide synthases: regulation and function." Eur Heart J **33**(7): 829-837, 837a-837d.
- Jin, X., Z. F. Yu, F. Chen, G. X. Lu, X. Y. Ding, L. J. Xie and J. T. Sun (2017). "Neuronal Nitric Oxide Synthase in Neural Stem Cells Induces Neuronal Fate Commitment via the Inhibition of Histone Deacetylase 2." Front Cell Neurosci **11**: 66.
- Madabhushi, R., F. Gao, A. R. Pfenning, L. Pan, S. Yamakawa, J. Seo, R. Rueda, T. X. Phan, H. Yamakawa, P. C. Pao, R. T. Stott, E. Gjoneska, A. Nott, S. Cho, M. Kellis and L. H. Tsai (2015). "Activity-Induced DNA Breaks Govern the Expression of Neuronal Early-Response Genes." Cell **161**(7): 1592-1605.
- Sii-Felice, K., V. Barroca, O. Etienne, L. Riou, F. Hoffschir, P. Fouchet, F. D. Boussin and M. A. Mouthon (2008). "Role of Fanconi DNA repair pathway in neural stem cell homeostasis." Cell Cycle **7**(13): 1911-1915.
- Sii-Felice, K., O. Etienne, F. Hoffschir, C. Mathieu, L. Riou, V. Barroca, C. Haton, F. Arwert, P. Fouchet, D. Boussin Fç and M. Mouthon (2008). "Fanconi DNA repair pathway is required for survival and long-term maintenance of neural progenitors." EMBO J **27**(5): 770-781.
- Wei, P. C., A. N. Chang, J. Kao, Z. Du, R. M. Meyers, F. W. Alt and B. Schwer (2016). "Long Neural Genes Harbor Recurrent DNA Break Clusters in Neural Stem/Progenitor Cells." Cell **164**(4): 644-655.

Reviewers' comments:

Reviewer #1 (Remarks to the Author):

In my initial review of this manuscript, I expressed concerns that the observations in the Ncf1^{-/-} cells may reflect redox regulation targeting multiple cellular proteins/pathways and the connection to DNA damage may not be central to this process. Thus, the phenomena observed in cells is not reflective of a control process determining neural progenitor fate in vivo. In the revision the authors have offered additional data to support their claims. In response to some of my comments they have presented in vivo data of the brain of Igfbp2^{-/-} and Ncf1^{-/-} mice (Suppl. Fig. 9). However, these data are not convincing or clear. The gamma-H2AX immunostaining is strange. The images show very high gamma-H2AX levels with what seems to be high background in the non-replicating region of the image. Even in WT, the green staining (gamma-H2AX levels) is very high/abundant and pan-nuclear. This is not the situation in a WT 12-week-old mouse brain. I also wonder about the specificity of the RAD51 staining as this known to not be a very robust reagent in mouse tissue (also the tissue section of the Igfbp2^{-/-} looks different to the other sections). Overall, this isn't very convincing in vivo support for the phenomena they are reporting.

Reviewer #2 (Remarks to the Author):

The authors have elegantly and successfully addressed all the concerns raised by this reviewer and the manuscript is now much improved.

Juan P Bolanos

Reviewer #3 (Remarks to the Author):

Review Shahin et al. "Redox-Dependent Igfbp2 Signaling Controls Brca1 DNA Damage Response to Govern Neural Stem Cell Fate"

The revised manuscript is improved and has addressed several of my concerns.

There are, however, some remaining issues:

1. The manuscript largely presents in vitro evidence for a function of the NCF1/ROS/IGFBP2/BRCA1 pathway in regulation of neural stem cell function. The central in

vitro findings, i.e., increased proliferation of stem cells and fate switch from neurons to oligodendrocytes are still not sufficiently addressed in vivo. The authors present some stainings and quantifications of oligodendrocyte markers and neuronal markers in the olfactory bulb of different KO mice. However, these data tell very little about potential fate switches of neural stem cells. Firstly, the birthdate of oligodendrocytes is unclear. Those oligos could be developmentally derived and may not be related to the adult NSCs that are under investigation. Secondly, given the literature that adult SVZ derived oligos largely remain close to the SVZ and that there is little evidence that adult generated oligos migrate towards the OB, it is necessary to show that these oligos are derived from NSCs in the SVZ that under physiological circumstances give rise to neurons. The minimum experiment would be that the authors birthdate neural stem cell progeny ideally via retroviruses in the SVZ and determine the neuron vs. oligo fate choice. BrdU-based fate mappings would be less ideal (as the origin of the oligos would be unclear) but still acceptable.

2. The authors should tone down statements about self-renewal and commitment and rewrite respective sections of the manuscript. In the literature, the classical hallmark of an NSC and for self-renewal is the formation of a neurosphere, which upon passaging generates again neurospheres. The authors do not perform any self-renewal experiments and interpret the formation of highly proliferative sheet like structures as self-renewal. In my view the correct representation of the data would be that the KO of NCF1 etc. results in altered proliferation behavior and an altered ability to form neurospheres. Please correct.

3. I do appreciate that the authors addressed the concern of the other reviewer regarding nNOS but for the reader the manuscript is now very confusing and the logic for performing / presenting experiments regarding nNOS function is unclear. For the sake of clarity I would strongly recommend to rewrite the relevant sections.

Responses to Reviewers' Comments

We greatly appreciate the reviewers' thoughtful comments and questions regarding our manuscript. We believe that addressing these concerns has greatly strengthened the manuscript, enabling us to draw stronger conclusions about the redox-dependent regulation of Igfbp2 by NADPH oxidase, and the involvement of the Brca1 DNA repair pathway in controlling cell fate decisions of neural stem cells (NSCs). We have added extensive additional data: several panels to Figure 3, supplementary figures 3, 8 and 9 that strengthen our original findings.

Details of the revisions are provided below in the point-by-point responses to both the editor's recommendations and the reviewers' comments. The reviewers' queries/comments are marked in italics by (Q) and responses are marked by (R) in non-italics font. Revisions to the manuscript are marked in the text by colored font to assist the reviewers in seeing the major edits.

Reviewers' comments

Reviewer #1 (Remarks to the Author):

Q1. In my initial review of this manuscript, I expressed concerns that the observations in the Ncf1^{-/-} cells may reflect redox regulation targeting multiple cellular proteins/pathways and the connection to DNA damage may not be central to this process. Thus, the phenomena observed in cells is not reflective of a control process determining neural progenitor fate in vivo.

R1. We appreciate the referee's concern about the specificity downstream of redox regulation. We have highlighted key findings that we believe support the specificity of the pathway studied in the above section addressed to the editor. We have rephrased several of these points below.

- 1) Reconstitution experiments with purified Igfbp2 (Fig. 2c–d) and cysteine mutants (Fig. 4c–d) demonstrate that oxidation of a single cysteine is required for neurosphere formation by *Igfbp2^{-/-}* and *Ncf1^{-/-}* NSCs. If this process was non-specifically controlled by other ROS mediated processes, how could this result be explained?
- 2) *Igfbp2^{-/-}* mice do not have altered ROS (Supplementary Fig. 6), yet they phenocopy *Ncf1^{-/-}* NSCs in terms of their self-renewing status and lack of commitment to form neurospheres (Fig. 2e–f). Thus, how can this reflect redox regulation targeting multiple cellular proteins/pathways and the connection to DNA damage? Both *Ncf1^{-/-}* and *Igfbp2^{-/-}* mice also have the same phenotype in the SVZ: 1) reduced histone γ -H2ax and increased Rad51 expression (Supplementary Fig. 9a–c) and 2) enhanced abundance of nascent newly born oligodendrocytes in the olfactory bulb (Supplementary Fig. 3a–f). This is a similar phenotype to differentiated *Ncf1^{-/-}* neurospheres (Supplementary Fig. 3g–k), which is reversed by inhibiting *Fanca*, *Fancd2* or *Rad51* (Supplementary Fig. 3l–m).
- 3) Altering the redox status of *Igfbp2^{-/-}* NSCs by H₂O₂ treatment did not restore neurosphere formation (Fig. 2g–h) nor reduce transcript levels of *Fanca*, *Fancd2* or *Rad51* to WT levels (Supplementary Fig. 7g–i). Both *Igfbp2^{-/-}* and *Ncf1^{-/-}* NSCs demonstrate the same alterations in elevated expression of *Fanca*, *Fancd2* and

- Rad51 DNA repair genes. Knockdown of these genes in *Igfbp2*^{-/-} and *Ncf1*^{-/-} NSCs similarly restores the ability of NSCs to commit to form neurospheres (Supplementary Fig. 8a–c), demonstrating the inhibition of DNA repair is the most distal consequence of *Igfbp2* action required for NSC lineage commitment.
- 4) We now also demonstrate that overexpression of *Fanca* in WT NSCs phenocopies *Ncf1*^{-/-} and *Igfbp2*^{-/-} NSCs (new data Fig. 3k–l), promoting formation of self-renewing sheets of NSCs while inhibiting neurosphere formation. This key result dissociates redox status from the mechanism and demonstrates that repression of DNA repair is the key event required for NSC commitment. Under these conditions, the *Ncf1*/*Igfbp2*/DNA repair axis remains intact, specifically demonstrating that NSC DNA repair status alone is solely responsible for NSC commitment and exit from a self-renewing state.

Thus, we have used biochemical reconstitution assays, loss of function, and gain of function assays to demonstrate specificity of the *Ncf1*/*Igfbp2*/DNA repair axis in the control of NSC self-renewal and lineage commitment events and have provided correlates to these events *in vivo*.

Q2. In the revision the authors have offered additional data to support their claims. In response to some of my comments they have presented in vivo data of the brain of Igfbp2-/- and Ncf1-/- mice (Suppl. Fig. 9). However, these data are not convincing or clear. The gamma-H2AX immunostaining is strange. The images show very high gamma-H2AX levels with what seems to be high background in the non-replicating region of the image. Even in WT, the green staining (gamma-H2AX levels) is very high/abundant and pan-nuclear. This is not the situation in a WT 12-week-old mouse brain. I also wonder about the specificity of the RAD51 staining as this known to not be a very robust reagent in mouse tissue (also the tissue section of the Igfbp2-/- looks different to the other sections). Overall, this isn't very convincing in vivo support for the phenomena they are reporting.

R2. We thank the reviewer for pointing out the pattern of γ -H2ax staining. We would like to refer to this paper¹. This was one of very few papers we could find with clear γ -H2ax staining of mouse brain. It shows γ -H2ax staining in both proliferating SVZ progenitors as well as post mitotic neurons. In fact, they mapped γ -H2ax⁺ neurons to the caudoputamen just next to the proliferating SVZ region. Thus, some neurons are positive for γ -H2ax staining. What we did differently was we used a pressure cooker to boil slides for epitope retrieval, which we stated in the methods section. Epitope retrieval might change and blur the focal pattern of γ -H2ax staining. The other thing that might have contributed to the non-focal pattern and high background of γ -H2ax staining, and for the higher number of Rad51 expressing cells, is that we used a wide field microscope DM6 (Leica) rather than a confocal microscope. For both γ -H2ax and Rad51 staining we did the non-primary antibody staining control. We used that control as a background to determine the γ -H2ax and Rad51 immune reactive nuclei.

Following the referee's comments, we have repeated the experiment and used a confocal laser scanning microscope LSM 980 (Zeiss). We also zoomed in on the SVZ to show γ -H2ax and Rad 51 expressing cells and show the single channel images. The new confocal images show the focal pattern of γ -H2ax and we have shown the same regions in all genotypes in a new Supplementary Fig. 9a–c replacing the old images and quantification. Again, we would like to thank the referee for bringing it up. We think these new confocal images have strengthened the manuscript.

We agree with the referee's comment that Rad51 is not a robust marker in WT SVZ. In our hands, the WT SVZ shows few Rad51⁺ nuclei, which appear to be mutually exclusive with γ -H2ax⁺ nuclei. However, *Ncf1*^{-/-} and *Igfbp2*^{-/-} SVZs show more Rad51⁺ and fewer γ -H2ax⁺ nuclei, which supports the model we are proposing. Now, new images using the confocal laser-scanning microscope of the same regions of the SVZ across the three genotypes show the extent of Rad51 expression in adult SVZ with more Rad51⁺ cells in the SVZ of *Ncf1*^{-/-} and *Igfbp2*^{-/-} compared to WT counterparts. We also did a non-primary antibody-stained control. We used that control to test the specificity of staining and as a background to determine Rad51 immune reactive cells. We also ran a 4-channel western blot analysis of NSC lysates of the three genotypes (Supplementary Fig. 9 d-h). Immunoblotting showed that *Ncf1*^{-/-} and *Igfbp2*^{-/-} NSCs have higher levels of Rad51 and Nestin but lower levels of γ -H2ax than WT counterparts. We used the same three primary antibodies we used for immunofluorescence labeling for Rad51, γ -H2ax, and Nestin in addition to Gapdh for normalization. As shown in Supplementary Fig. 9 d, all antibodies labelled a single band at the correct molecular weight for each target. This further confirms the specificity of the antibodies we used.

1. Barral S, Beltramo R, Salio C, Aimar P, Lossi L, Merighi A. Phosphorylation of histone H2AX in the mouse brain from development to senescence. *Int J Mol Sci* **15**, 1554-1573 (2014).

Reviewer #2 (Remarks to the Author):

The authors have elegantly and successfully addressed all the concerns raised by this reviewer and the manuscript is now much improved.

Juan P Bolanos

The authors would like to thank the second reviewer for his constructive comments.

Reviewer #3 (Remarks to the Author):

Review Shahin et al. "Redox-Dependent Igfbp2 Signaling Controls Brca1 DNA Damage Response to Govern Neural Stem Cell Fate"

The revised manuscript is improved and has addressed several of my concerns. There are, however, some remaining issues:

Q1. The manuscript largely presents in vitro evidence for a function of the NCF1/ROS/IGFBP2/BRCA1 pathway in regulation of neural stem cell function. The central in vitro findings, i.e., increased proliferation of stem cells and fate switch from neurons to oligodendrocytes are still not sufficiently addressed in vivo. The authors present some stainings and quantifications of oligodendrocyte markers and neuronal markers in the olfactory bulb of different KO mice. However, these data tell very little about potential fate switches of neural stem cells. Firstly, the birthdate of oligodendrocytes is unclear. Those oligos could be developmentally derived and may not be related to the adult NSCs that are under investigation. Secondly, given the literature that adult SVZ derived oligos largely remain close to the SVZ and that there is little evidence that adult generated oligos migrate towards the OB, it is necessary to show that these oligos are derived from NSCs in the SVZ that under physiological circumstances give rise to neurons. The minimum experiment would be that the authors

birthdate neural stem cell progeny ideally via retroviruses in the SVZ and determine the neuron vs. oligo fate choice. BrdU-based fate mappings would be less ideal (as the origin of the oligos would be unclear) but still acceptable.

R1. We appreciate the reviewer's constructive criticism and suggestion for linking our *in vitro* findings with *in vivo* relevance. We performed one of the proposed experiments and now show the results of EdU pulse labeling of mice in a new Supplementary Fig. 3a–f. We now show that *Ncf1*^{-/-} and *Igfbp2*^{-/-} mice form more new EdU⁺ oligodendrocytes and fewer EdU⁺ neurons in the olfactory bulb than WT counterparts.

Q2. The authors should tone down statements about self-renewal and commitment and rewrite respective sections of the manuscript. In the literature, the classical hallmark of an NSC and for self-renewal is the formation of a neurosphere, which upon passaging generates again neurospheres. The authors do not perform any self-renewal experiments and interpret the formation of highly proliferative sheet like structures as self-renewal. In my view the correct representation of the data would be that the KO of NCF1 etc. results in altered proliferation behavior and an altered ability to form neurospheres. Please correct.

R2. We are grateful that the referee pointed out this limitation and agree with this definition of self-renewing NSCs. We have now performed serial passage experiments (Supplemental Fig. 8d-h) where we repeatedly treated WT, *Ncf1*^{-/-} and *Igfbp2*^{-/-} NSCs with negative control dsiRNA or dsiRNA against Rad51 for three consecutive passages (P1–3). *Ncf1*^{-/-} and *Igfbp2*^{-/-} NSCs formed neurospheres at every passage when treated with Rad51 dsiRNA, while the control dsiRNA-treated culture did not form neurospheres. WT NSCs formed neurospheres in all conditions at each passage. This supports the notion that all genotypes (WT, *Ncf1*^{-/-} and *Igfbp2*^{-/-}) contain self-renewing NSCs but differ in their capacity to commit to form neurospheres based their differing levels of Rad51. We have also rewritten the text such that proliferation is emphasized prior to presenting this experiment that addressed self-renewal.

Q3. I do appreciate that the authors addressed the concern of the other reviewer regarding nNOS but for the reader the manuscript is now very confusing and the logic for performing / presenting experiments regarding nNOS function is unclear. For the sake of clarity I would strongly recommend to rewrite the relevant sections.

R3. We appreciate the reviewer's valuable comment. We have rewritten the relevant sections of the manuscript to improve the logic for performing these experiments and have consolidated these comments to one major section for clarity.

REVIEWERS' COMMENTS

Reviewer #3 (Remarks to the Author):

The authors have adequately addressed my remaining concerns. In particular the new in vivo data strengthen the central claim of the manuscript. Moreover the manuscript is written in a clearer way and is much better understandable.

Reviewer #4 (Remarks to the Author):

The authors have successfully addressed the concern of whether DNA damage repair is central to the mechanism regulating neural stem cell fate, especially through the new Fanca overexpression experiment. In addition, the new confocal imaging shows the expected pattern of Gamma-H2AX. I recommend the publication of the manuscript in Nature Communications.